

# Fusion and monodromy in the Temperley-Lieb category

**Jonathan Belletête** [1][*] **and Yvan Saint-Aubin**[2][†]

**1** Institut de Physique Théorique, Université Paris Saclay,
CEA, CNRS, 91191 Gif Sur Yvette, France
**2** Département de mathématiques et de statistique,
Université de Montréal, Montréal, QC, Canada, H3C 3J7

[*] jonathan.belletete@ipht.fr, [†] yvan.saint-aubin@umontreal.ca

## Abstract

Graham and Lehrer (1998) introduced a Temperley-Lieb category $\widetilde{\mathsf{TL}}$ whose objects are the non-negative integers and the morphisms in $\mathrm{Hom}(n, m)$ are the link diagrams from $n$ to $m$ nodes. The Temperley-Lieb algebra $\mathsf{TL}_n$ is identified with $\mathrm{Hom}(n, n)$. The category $\widetilde{\mathsf{TL}}$ is shown to be monoidal. We show that it is also a braided category by constructing explicitly a commutor. A twist is also defined on $\widetilde{\mathsf{TL}}$. We introduce a module category $\mathrm{Mod}_{\widetilde{\mathsf{TL}}}$ whose objects are functors from $\widetilde{\mathsf{TL}}$ to $\mathrm{Vect}_{\mathbb{C}}$ and define on it a fusion bifunctor extending the one introduced by Read and Saleur (2007). We use the natural morphisms constructed for $\widetilde{\mathsf{TL}}$ to induce the structure of a ribbon category on $\mathrm{Mod}_{\widetilde{\mathsf{TL}}}(\beta = -q - q^{-1})$, when $q$ is not a root of unity. We discuss how the braiding on $\widetilde{\mathsf{TL}}$ and integrability of statistical models are related. The extension of these structures to the family of dilute Temperley-Lieb algebras is also discussed.



## 1  Introduction

The (original) family of the Temperley-Lieb algebras was cast into a categorical framework by Graham and Lehrer [1] in 1998. A decade later Read and Saleur [2] introduced a product $-_1 \times_f -_2$ between two modules over two (maybe distinct) Temperley-Lieb algebras. Still later this product was computed between several families of modules by Gainutdinov and Vasseur [3], and Bellètête [4]. Their recent results (obtained in 2012 and 2015 respectively) lead to natural questions: how can one define the module category over Graham and Lehrer's category? Does the natural braiding that exists on Graham and Lehrer's category (described for example by Turaev [5]) extend to this module category? And how many of the defining properties of tensor categories does the module category satisfy? The present paper answers these questions.

Statistical models in two dimensions are defined by an evolution operator or a transfer matrix acting on finite-dimensional vector spaces. The sizes of both the lattice and the vector spaces are parameters of the formulation. The limit when these parameters go to infinity is known in some cases (numerically or rigorously) to be a conformal field theory. For several XXZ and loop models [6,7], the Hamiltonian is first defined as an element of a Temperley-Lieb algebra $\mathsf{TL}_n$, or one of its generalizations, and the actual linear operator is obtained as the representative of this element in some representations over the algebra. The fusion product is an algebraic construction, actually a bifunctor, that associates to two modules M and N over $\mathsf{TL}_m$ and $\mathsf{TL}_n$ respectively a module over $\mathsf{TL}_{m+n}$. As said above, for the Temperley-Lieb algebra, such a fusion product $-_1 \times_f -_2$ was introduced by Read and Saleur [2] and computed in many cases by Gainutdinov and Vasseur [3] and Bellètête [4]. It is associative and commutative, and the braiding gives the isomorphism between M $\times_f$ N and N $\times_f$ M.

There are reasons to believe that algebraic information obtained from the finite algebras, either the Temperley-Lieb family, its dilute counterpart or any other one, is intimately related to analogous structures of the CFTs and should help understand them. First there is compelling evidence that, in the limit when the size of the lattice goes to infinity, the spectrum of the Hamiltonian, properly scaled, reproduces characters of the Virasoro algebra. Second, in some representations of the $\mathsf{TL}$ family, the Hamiltonian has Jordan blocks (of size $2 \times 2$) [7, 8], indicating a possible link to logarithmic CFTs. Third, when restricted to $\mathsf{TL}$-modules that are known to give rise to the Virasoro modules appearing in minimal CFTs, the highly non-trivial fusion product defined between $\mathsf{TL}$ modules does reproduce the simple fusion rules of these minimal CFTs. Since the operator product expansion of CFTs leads to a fusion product between modules over the Virasoro algebra whose many properties are captured into a tensor category, it is natural to ask how many of these properties are shared by Read's and Saleur's fusion.

The (original) family of the Temperley-Lieb algebras was cast into a categorical framework by Graham and Lehrer [1] while they were actually studying another family, the periodic (or affine) Temperley-Lieb algebras. The construction brings together all algebras $\mathsf{TL}_n(\beta), n \geq 0$, in the same category $\widetilde{\mathsf{TL}}(\beta)$. Their formulation will be the starting point of 2 where the requirements for a category to be monoidal and then braided will be fulfilled for the $\mathsf{TL}$ family.

Even though the braiding on $\widetilde{\mathsf{TL}}(\beta)$ is already known, the twist presented in this section is new to our knowledge.

Section 3 defines a module category $\mathrm{Mod}_{\widetilde{\mathsf{TL}}}$ whose objects are functors from $\widetilde{\mathsf{TL}}$ to $\mathrm{Vect}_{\mathbb{C}}$. The associator, commutor and twist defined on $\widetilde{\mathsf{TL}}$ are shown to induce similar natural transformations on $\mathrm{Mod}_{\widetilde{\mathsf{TL}}}$. 4 then shows how the integrability of two-dimensional statistical models and the components of the commutor $\eta_{r,s}$ defining the braiding are related. 5 extends the results to the family of dilute Temperley-Lieb algebras $\mathrm{dTL}_n$. A short conclusion follows.

## 2 The Temperley-Lieb category

Graham and Lehrer [1] showed that the algebras $\mathsf{TL}_n(\beta)$, $n \geq 0$, can be studied as a whole and given the structure of a category $\widetilde{\mathsf{TL}}$. The goal of this section is to recall the definitions of monoidal and braided categories and show that the Temperley-Lieb category $\widetilde{\mathsf{TL}}$ is braided. Some of the data necessary to define a braided category are known for the $\mathsf{TL}$ family (see for example [5]). However giving the details here fulfills several goals. It provides a pedagogical introduction to these structures with detailed proofs. It also establishes many of their properties that will play a crucial role in Section 3.

### 2.1 $\widetilde{\mathsf{TL}}(\beta)$ as a monoidal category

The first step is to cast the family of algebras $\mathsf{TL}_n, n \geq 0$, into a category and show that the additional requirements of a monoidal category are easily fulfilled.

*We take the convention that morphisms and functors acts on the* left, *so that* $(FG)(X) \equiv F(G(X))$.

The *Temperley-Lieb category* $\widetilde{\mathsf{TL}}$ is defined as follows. The objects of the category are the non-negative integers:
$$\mathrm{Ob}\,\widetilde{\mathsf{TL}} = \mathbb{N}_0 = \{0, 1, 2, \ldots\}.$$

The sets of morphisms $\mathrm{Hom}(n, m)$ from $n$ to $m$ is empty if $n$ and $m$ do not have the same parity and, if they do, are defined as the sets of formal $\mathbb{C}$-linear combinations of $(m, n)$-diagrams. A $(m, n)$-diagram $\alpha \in \mathrm{Hom}(n, m)$ is composed of two vertical columns of $m$ nodes on the left, and $n$ nodes on the right, linked pairwise by non-intersecting strings. For instance, here are a pair of $(4, 2)$-diagrams and one $(2, 4)$-diagram:

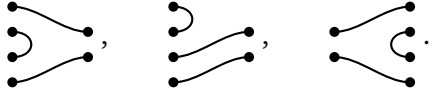

Note that the third diagram can be obtained from the first by reflection through a vertical line midway between the two columns of nodes. We will call the result of this reflection the *transpose* of the diagram. The identity morphism $1_n \in \mathrm{Hom}(n, n)$ is the $(n, n)$-diagram where every point on the left is linked to the one at the same height on the right. (The identity morphism $1_0 \in \mathrm{Hom}(0, 0)$ exists (by definition), but it represented graphically by an empty space.) Compositions of morphisms are defined by linearly expanding the composition rules for diagrams. For an $(m, n)$-diagram $b$ and a $(k, m)$-diagram $c$, the composition $c \circ b$ is a $(k, n)$-diagram defined by first putting $c$ on the left of $b$, identifying the $m$ points on the neighboring sites, joining the strings that meets there, and then removing these $m$ nodes. If there is a string no longer attached to any points, it is removed and replaced by a factor $\beta \in \mathbb{C}$. Here is an example of the composition of $(4, 2)$- and a $(2, 4)$-diagrams:

$$\left( \vcenter{\hbox{\includegraphics}} \right) \circ \left( \vcenter{\hbox{\includegraphics}} \right) \equiv \vcenter{\hbox{\includegraphics}} \equiv \vcenter{\hbox{\includegraphics}}$$

and of the same diagrams in the other order:

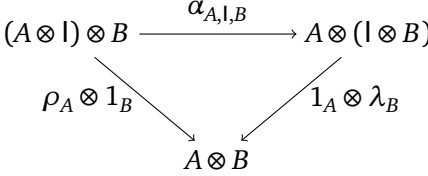

The associativity of the composition of diagrams is easily verified. The depiction of $1_0$ by a simple space is consistent with the depiction of the composition by concatenation of diagrams. For example the following product of the $(2,0)$-diagram $d$ and $(0,2)$-diagram $e$

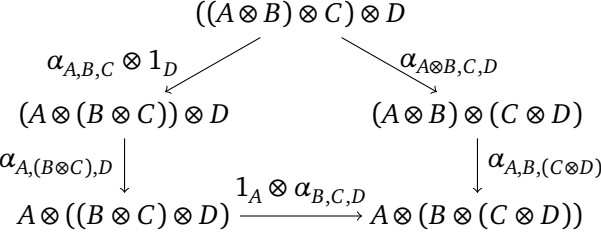

could equally be understood as $d \circ 1_0 \circ e$. Note finally that $\mathrm{End}(n) \equiv \mathsf{TL}_n(\beta)$ is the usual Temperley-Lieb algebra $\mathsf{TL}_n(\beta)$. The category $\widetilde{\mathsf{TL}}$ can be easily enriched to become a monoidal one.

A category $\mathcal{C}$ is said to be *monoidal* if it is equipped with the following structures [9–11]:

1. A bifunctor $-\otimes- : \mathcal{C} \times \mathcal{C} \longrightarrow \mathcal{C}$, called the *tensor product*;

2. An object $\mathsf{I} \in \mathrm{Ob}(\mathcal{C})$ called the *identity*;

3. Three natural isomorphisms[1]:

   - $\alpha : (-_1 \otimes -_2) \otimes -_3 \longrightarrow -_1 \otimes (-_2 \otimes -_3)$, called the *associator*.
   - $\lambda : \mathsf{I} \otimes - \longrightarrow -$, the *left unitor*.
   - $\rho : - \otimes \mathsf{I} \longrightarrow -$, the *right unitor*.

Moreover these structures have to satisfy the *triangle* and the *pentagon axioms*. These axioms require that the diagrams in Figures 1 and 2 commute for all $A, B, C, D \in \mathrm{Ob}\,\mathcal{C}$. Finally, if the associator, the left and right unitors are all identity isomorphisms, the category is said to be *strict*.

$$(A \otimes \mathsf{I}) \otimes B \xrightarrow{\ \alpha_{A,\mathsf{I},B}\ } A \otimes (\mathsf{I} \otimes B)$$

$$\rho_A \otimes 1_B \searrow \qquad \swarrow 1_A \otimes \lambda_B$$

$$A \otimes B$$

Figure 1: The triangle diagram

$$((A \otimes B) \otimes C) \otimes D$$

$$\alpha_{A,B,C} \otimes 1_D \swarrow \qquad \searrow \alpha_{A \otimes B, C, D}$$

$$(A \otimes (B \otimes C)) \otimes D \qquad (A \otimes B) \otimes (C \otimes D)$$

$$\alpha_{A,(B \otimes C),D} \downarrow \qquad\qquad \downarrow \alpha_{A,B,(C \otimes D)}$$

$$A \otimes ((B \otimes C) \otimes D) \xrightarrow{\ 1_A \otimes \alpha_{B,C,D}\ } A \otimes (B \otimes (C \otimes D))$$

Figure 2: The pentagon diagram

---

[1]We recall that a *natural isomorphism* $\mu : F \to G$ between two functors $F, G : \mathcal{C}_1 \to \mathcal{C}_2$ associates to each $A \in \mathrm{Ob}(\mathcal{C}_1)$ an invertible morphism $\mu_A \in \mathrm{Hom}_{\mathcal{C}_2}(F(A), G(A))$ such that $\mu_B \circ F(f) = G(f) \circ \mu_A$, for all $f \in \mathrm{Hom}_{\mathcal{C}_1}(A, B)$. The morphism $\mu_A$ is called the *component* of $\mu$ at $A$.

**Definition 1.** *Let $\mathcal{C}$ be the Temperley-Lieb category $\widetilde{\mathsf{TL}}$. Define the bifunctor $-\otimes-$ in the following way. For objects $n, m \in \mathrm{Ob}\,\widetilde{\mathsf{TL}}$, simply set* [2] *$n \otimes m \equiv n + m$ where "+" stands for the addition in $\mathbb{N}_0$ and, thus, the identity object is $\mathsf{I} = 0 \in \mathrm{Ob}\,\widetilde{\mathsf{TL}}$. For a $(k, n)$-diagram $b$ and a $(t, m)$-diagram $c$, the $(k + t, n + m)$-diagram $b \otimes c$ is obtained by simply drawing $b$ on top of $c$. For example, taking $b, c$ as in the previous example gives*

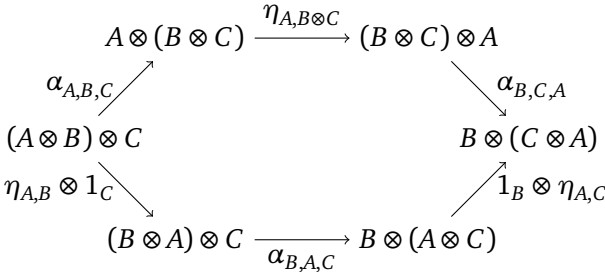

*This is then expanded bilinearly to all morphisms. The associator $\alpha_{m,n,k}$ is the isomorphism $(m + n) + k \mapsto m + (n + k)$ and the unitors are $0 + m \mapsto m$ and $m + 0 \mapsto m$ respectively.*

Since $(\mathbb{N}_0, +)$ is a monoid, the axioms are trivially verified for the objects. It is easy to verify that the axioms also hold for the morphisms and, thus, $\widetilde{\mathsf{TL}}$ is a strict monoidal category.

## 2.2 $\widetilde{\mathsf{TL}}(\beta)$ as a braided category

Let $\mathcal{C}$ be a monoidal category and let the *opposite tensor product* between two objects $A, B \in \mathrm{Ob}\,\mathcal{C}$ be defined as $A \otimes^{\mathrm{op}} B \equiv B \otimes A$. The category $\mathcal{C}$ is *braided* if there is a natural isomorphism $\eta : -\otimes- \to -\otimes^{\mathrm{op}}-$ such that the two *hexagon* diagrams in Figures 3 and 4 commute for all $A, B, C \in \mathrm{Ob}\,\mathcal{C}$. When such a natural isomorphism exists, it is called a *commutor*. If, for all $A, B \in \mathrm{Ob}\,\mathcal{C}$, $\eta_{A,B} \circ \eta_{B,A} = 1_{B \otimes A}$, the category is said to be *symmetric*.

$$
\begin{array}{ccc}
& A \otimes (B \otimes C) \xrightarrow{\;\eta_{A,B\otimes C}\;} (B \otimes C) \otimes A & \\
{\scriptstyle \alpha_{A,B,C}}\nearrow & & \searrow{\scriptstyle \alpha_{B,C,A}} \\
(A \otimes B) \otimes C & & B \otimes (C \otimes A) \\
{\scriptstyle \eta_{A,B} \otimes 1_C}\searrow & & \nearrow{\scriptstyle 1_B \otimes \eta_{A,C}} \\
& (B \otimes A) \otimes C \xrightarrow[\;\alpha_{B,A,C}\;]{} B \otimes (A \otimes C) &
\end{array}
$$

Figure 3: The first hexagon diagram

In a strict braided category, the hexagon diagrams are equivalent to the two following identities:

$$\eta_{A,B\otimes C} = \left(1_B \otimes \eta_{A,C}\right) \circ \left(\eta_{A,B} \otimes 1_C\right), \tag{1}$$

$$\eta_{B\otimes C,A} = \left(\eta_{B,A} \otimes 1_C\right) \circ \left(1_B \otimes \eta_{C,A}\right). \tag{2}$$

To endow $\widetilde{\mathsf{TL}}$ with a braiding requires more work than to define its monoidal structure. We start by outlining the strategy. Since $\widetilde{\mathsf{TL}}$ is strict, the hexagon diagrams are equivalent to

$$\eta_{n,m+k} = \left(1_m \otimes \eta_{n,k}\right) \circ \left(\eta_{n,m} \otimes 1_k\right), \tag{3}$$

---

[2] The category $\widetilde{\mathsf{TL}}$ is not additive. Indeed one of the requirements for additivity is the existence of direct sum objects for any finite set of objects. The direct sum of two objects $m, n \in \mathrm{Ob}\,\widetilde{\mathsf{TL}}$ would be given by an object $(m \oplus n) \in \mathrm{Ob}\,\widetilde{\mathsf{TL}}$ together with maps $m \xrightarrow{q_m} (m \oplus n)$ and $n \xrightarrow{q_n} (m \oplus n)$ satisfying some universal property. However if $m$ and $n$ are of different parity, one of the two sets $\mathrm{Hom}(m, (m \oplus n))$ and $\mathrm{Hom}(n, (m \oplus n))$ is empty and the basic requirements for the existence of the direct sum cannot be met.

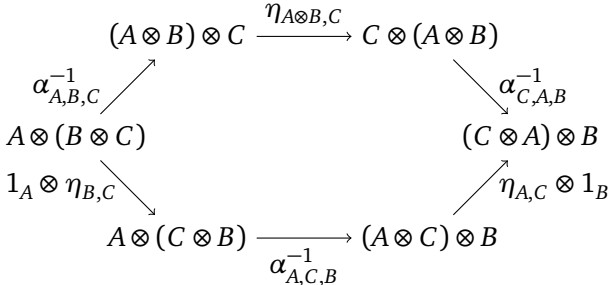

Figure 4: The second hexagon diagram

$$\eta_{u+v,w} = \left(\eta_{u,w} \otimes 1_v\right) \circ \left(1_u \otimes \eta_{v,w}\right). \tag{4}$$

It follows that, if we can find $\eta_{1,1}$, the other $\eta_{n,m}$, $n, m \geq 1$, will be uniquely defined by these two conditions, provided that they are consistent, that is, if $\eta_{n,m}$ satisfy the above two conditions, then so does $\eta_{n+1,m} \equiv \left(\eta_{n,m} \otimes 1_1\right) \circ \left(1_n \otimes \eta_{1,m}\right)$, for instance. Proposition 2.1 will establish this consistency. We shall then build the $\eta_{n,m}$ recursively. It will then remain to prove that these $\eta_{m,n}$ define natural isomorphisms. This will require several steps: Lemma 2.2 will express the morphisms $\eta_{r,s}$ in terms of $\eta_{1,1}$ only and a short computation will express $\eta_{1,1}$ in terms of the generators $e_i$ of Temperley-Lieb algebras. Lemmas 2.3 to 2.5 show how the $\eta_{r,s}$ braid with the $e_i$ and some diagrams in $\text{Hom}(n, 0)$ and $\text{Hom}(0, n)$. Then Proposition 2.6 proves that the $\eta_{r,s}$ form together a commutor for the category $\widetilde{\mathsf{TL}}$.

The hexagon axioms fix the isomorphisms $\eta_{n,0}$ and $\eta_{0,w}$. Indeed, when all integers are set to 0, (3) gives $\eta_{0,0} = (1_0 \otimes \eta_{0,0}) \circ (\eta_{0,0} \otimes 1_0)$ and thus $\eta_{0,0} = 1_0$. Similarly the same equation for $\eta_{n,0+1}$ leads to $\eta_{n,0} = 1_n$. Hence $\eta_{n,0} = \eta_{0,n} = 1_n$ for all $n \geq 0$.

**Proposition 2.1.** *If the morphisms $\{\eta_{i,j}\}_{0 \leq i \leq r, 0 \leq j \leq s}$ satisfy equations (3) and (4) for all $0 \leq n, u + v \leq r$ and $0 \leq m + k, w \leq s$, then so do $\eta_{r+1,s}$ and $\eta_{r,s+1}$ defined as*

$$\eta_{r+1,s} \equiv \left(\eta_{r,s} \otimes 1_1\right) \circ \left(1_r \otimes \eta_{1,s}\right) \qquad and \qquad \eta_{r,1+s} \equiv \left(1_1 \otimes \eta_{r,s}\right) \circ \left(\eta_{r,1} \otimes 1_s\right). \tag{5}$$

*Proof.* We only give the proof for $\eta_{r+1,s}$ and equation (4), as the other checks are similar. Suppose that $n + m = r + 1$ with $0 \leq n \leq r$ and $1 \leq m \leq s$. The steps are the following:

$$
\begin{aligned}
\left(\eta_{n,s} \otimes 1_m\right) \circ \left(1_n \otimes \eta_{m,s}\right) &\overset{1}{=} \left(\eta_{n,s} \otimes 1_m\right) \circ \left(1_n \otimes \eta_{m-1,s} \otimes 1_1\right) \circ \left(1_n \otimes 1_{m-1} \otimes \eta_{1,s}\right) \\
&\overset{2}{=} \left(\eta_{n,s} \otimes 1_{m-1} \otimes 1_1\right) \circ \left(1_n \otimes \eta_{m-1,s} \otimes 1_1\right) \circ \left(1_{n+m-1=r} \otimes \eta_{1,s}\right) \\
&\overset{3}{=} \left(\left(\left(\eta_{n,s} \otimes 1_{m-1}\right) \circ \left(1_n \otimes \eta_{m-1,s}\right)\right) \otimes 1_1\right) \circ \left(1_r \otimes \eta_{1,s}\right) \\
&\overset{4}{=} \left(\eta_{r,s} \otimes 1_1\right) \circ \left(1_r \otimes \eta_{1,s}\right) \\
&\overset{5}{=} \eta_{r+1,s}.
\end{aligned}
$$

Steps 1 and 4 are obtained by using the fact that $\eta_{m,s}$ and $\eta_{r,s}$ satisfy the hexagon identity (4). Steps 2 and 3 use the property $1_i \otimes 1_j = 1_{i+j}$ of identity morphisms that holds for all non-negative integers $i, j$. Finally step 5 is the proposed definition of $\eta_{r+1,s}$. $\qquad\square$

The next lemma solves the recursive expressions (5) in terms of the "elementary component" $\eta_{1,1}$ only.

**Lemma 2.2.** *The morphisms $\eta_{r,s}$, with $r, s \geq 1$, satisfy equations* (3) *and* (4) *if and only if they are given by*

$$\eta_{r,s} = \prod_{i=1}^{s} \left( \prod_{j=r-1}^{0} t_{i+j}(r+s) \right) = \prod_{i=r}^{1} \left( \prod_{j=0}^{s-1} t_{i+j}(r+s) \right), \tag{6}$$

*where $t_i(n) \equiv 1_{i-1} \otimes \eta_{1,1} \otimes 1_{n-i-1} \in \mathrm{Hom}(n, n)$ and the factors in a product are listed starting from the right, that is, $\prod_{i=1}^{s} t_i = t_s t_{s-1} \ldots t_2 t_1$ and $\prod_{i=s}^{1} t_i \equiv t_1 t_2 \ldots t_{s-1} t_s$.*

*Proof.* The proof of the first part is obtained by induction on $r$ and $s$. Taking the induction first on $r$, then on $s$ gives the first expression, while doing the inductions in the reverse order yields the second. The proof of the former is given as example. When $r = s = 1$, the first expression is simply $\eta_{1,1}$ and the statement is trivially true. Assume therefore that the result stands for $\eta_{r,1}$. If $\eta_{r+1,1}$ satisfies equation (4), then in particular

$$\eta_{r+1,1} = \left( \eta_{r,1} \otimes 1_1 \right) \circ \underbrace{\left( 1_r \otimes \eta_{1,1} \right)}_{t_{r+1}(r+2)},$$

which is $\eta_{r+1,1}$ as given by the first expression in (6). Assume then that the result stands for some $r, s \geq 1$. Then (3) gives

$$\eta_{r,s+1} = \left( 1_s \otimes \eta_{r,1} \right) \circ \left( \eta_{r,s} \otimes 1_1 \right)$$
$$= \left( \prod_{j=r-1}^{0} t_{j+s+1}(r+s+1) \right) \circ \prod_{i=1}^{s} \left( \prod_{j=r-1}^{0} t_{i+j}(r+s+1) \right),$$

which is the expression for $\eta_{r,s+1}$ given in (6).

The converse can be obtained as follows. The first expression gives

$$(1_m \otimes \eta_{n,k}) \circ (\eta_{n,m} \otimes 1_k) = \left( \prod_{i=1}^{k} \prod_{j=n-1}^{0} t_{i+j+m} \right) \circ \left( \prod_{i=1}^{m} \prod_{j=n-1}^{0} t_{i+j} \right)$$
$$= \left( \prod_{i=m+1}^{m+k} \prod_{j=n-1}^{0} t_{i+j} \right) \circ \left( \prod_{i=1}^{m} \prod_{j=n-1}^{0} t_{i+j} \right)$$
$$= \left( \prod_{i=1}^{m+k} \prod_{j=n-1}^{0} t_{i+j} \right) = \eta_{n,m+k}$$

and, thus, satisfies (3). The second expression is shown similarly to satify (4). The proof that the first expression satisfies (4) is harder and it is then easier, though tedious, to prove that the two expressions are equal. It is done using the identity $t_i t_j = t_j t_i$ for $|i - j| > 1$, that follows from the definition of the $t_i(n)$. Here is an example. The two expressions for $\eta_{2,3}$ are $(t_3 t_4)(t_2 t_3)(t_1 t_2)$ and $(t_3 t_2 t_1)(t_4 t_3 t_2)$ and those for $\eta_{2,2}$ are $(t_2 t_3)(t_1 t_2)$ and $(t_2 t_1)(t_3 t_2)$. Assuming that the latter are equal, the former are shown to be equal by

$$(t_3 t_4)\big((t_2 t_3)(t_1 t_2)\big) = (t_3 t_4)\big((t_2 t_1)(t_3 t_2)\big) = (t_3 t_2 t_1)(t_4 t_3 t_2)$$

where the two expressions for $\eta_{2,2}$ gives the first equatlity while the commutativity of $t_4$ with $t_2$ and $t_1$ gives the second. The argument can be extended into a proof by induction on the sum $r + s$ of the indices of $\eta_{r,s}$. □

The next step is to find an expression for $\eta_{1,1}$. Since $\eta_{1,1} : 1 \otimes 1 \to 1 \otimes 1$ is an element of $\text{End}(2) \simeq \mathsf{TL}_2(\beta)$, which is two-dimensional, there exists $\alpha, \gamma \in \mathbb{C}$ such that $\eta_{1,1} = \alpha 1_2 + \gamma e_1(2)$, where the notation

$$e_i(n) = 1_{i-1} \otimes \; \rangle \; \zeta \; \otimes 1_{n-(i+1)}$$

is used. It can be checked directly from this definition that the $e_i$ satisfy the Temperley-Lieb defining relations:

$$e_i(n)e_i(n) = \beta e_i(n), \qquad e_i(n)e_{i\pm 1}(n)e_i(n) = e_i(n), \tag{7}$$

$$e_i(n)e_j(n) = e_j(n)e_i(n), \qquad \text{if } |i - j| > 1. \tag{8}$$

In fact, it can be proved that the set $\{e_i(n)\}_{1 \le i < n}$ generates $\text{End}(n) = \mathsf{TL}_n(\beta)$. Using these relations, it can be seen that $\eta_{1,1}$ is invertible provided that $\alpha \ne 0$. Now, if the family of $\eta_{r,s}$ is to define a commutor then, in particular, it must verify

$$\eta_{1,2}e_2(3) = e_1(3)\eta_{1,2} \qquad \text{and} \qquad \eta_{1,2}(1_1 \otimes z) = (z \otimes 1_1)\eta_{1,0} \tag{9}$$

where

$$z = \; \rangle \; \in \text{Hom}_{\widetilde{\mathsf{TL}}}(0, 2) \tag{10}$$

and $\eta_{1,2} = (1_1 \otimes \eta_{1,1}) \circ (\eta_{1,1} \otimes 1_1) = \alpha^2 1_3 + \alpha\gamma(e_1(3) + e_2(3)) + \gamma^2 e_2(3)e_1(3)$. The first equation of (9) will be satisfied if and only if $\alpha^2 + \beta\alpha\gamma + \gamma^2 = 0$, while the second will be if and only if $\alpha\gamma = 1$. Solving these equations yields

$$\alpha = \pm q^{\pm 1/2}, \qquad \gamma = 1/\alpha,$$

where $q \in \mathbb{C}^\times$ is such that $\beta = -q - q^{-1}$ and the two $\pm$ signs are independent. There are thus four solutions. Note that one of the $\pm$ is responsible for an overal sign on $\eta_{1,1}$ while the remaining one mirrors the invariance of $\beta$ under $q \mapsto q^{-1}$. Without loss of generalities, we shall concentrate on the following choice:

$$t_i(n) = q^{1/2}(1_n + q^{-1}e_i(n)) \qquad \text{and} \qquad t_i(n)^{-1} = q^{-1/2}(1_n + qe_i(n)) \tag{11}$$

and $\eta_{1,1} : 1 \otimes 1 \to 1 \otimes 1$ is simply $\eta_{1,1} = t_1(2)$. These building blocks $t_i(n)$ of the $\eta_{r,s}$ have appeared numerous times in the literature. The identity (16) below was recognized by Chow [12] as crucial to identify the center of braid groups. Much later Martin [13] used the $t_i$ (up to a factor) to construct central elements of the Temperley-Lieb algebra.

It can also be useful to introduce diagrams representing $t_i(n)$ and $t_i(n)^{-1}$; we choose the following

$$t_1(2) \equiv \; \asymp \; , \qquad t_1(2)^{-1} \equiv \; \asymp \; . \tag{12}$$

The other $t_i(n)$ can be built from these by using the tensor product of morphisms. These diagrams are concatenated using the same rules as for the other diagrams representing morphisms in the category, so diagrams with isotopic strings are equivalent. Note however that diagrams related through a Reidemeister move of type I are not necessarily equivalent; for instance,

$$t_1(2)e_1 \equiv \; \asymp \; \rangle \quad \zeta \; = -(q)^{-3/2}e_1 = -(q)^{-3/2} \; \rangle \quad \zeta \; . \tag{13}$$

The following lemmas will give the behaviour of these crossings under the Reidermeister moves of the two other types.

It now remains to show that this choice does defines a braiding on $\widetilde{\mathsf{TL}}$, but doing so requires a few lemmas. From now on, we shall omit the arguments specifying the Hom-space, unless

they are needed to avoid confusion, and assume that these arguments are large enough for the expressions to make sense. For example the next lemma proves that $t_i t_{i+1} e_i = e_{i+1} e_i$. The statement stands for $t_i(n) t_{i+1}(n) e_i(n) = e_{i+1}(n) e_i(n)$ for all $i + 2 \leq n$ as $t_{i+1}(n)$ and $e_{i+1}(n)$ act non-trivially on nodes $i+1$ and $i+2$ of the elements of $\mathrm{Hom}(n,n)$. The next three lemmas prepare the proof that the $\eta_{r,s}$'s are natural isomorphisms and thus define a braiding on $\widetilde{\mathsf{TL}}$. The first is obtained by direct computation.

**Lemma 2.3.** *The morphisms $t_i$ and $e_i$ satisfy*

$$t_i t_{i+1} e_i = e_{i+1} e_i = e_{i+1} t_i t_{i+1}, \tag{14}$$

$$t_{i+1} t_i e_{i+1} = e_i e_{i+1} = e_i t_{i+1} t_i, \tag{15}$$

$$t_i t_{i+1} t_i = t_{i+1} t_i t_{i+1}. \tag{16}$$

In terms of diagrams, this lemma can be written

$$t_1 t_2 e_1 \equiv \quad \cdots \quad \equiv e_2 t_1 t_2, \tag{17}$$

$$t_2 t_1 e_2 \equiv \quad \cdots \quad = e_1 t_2 t_1, \tag{18}$$

$$t_1 t_2 t_1 = \quad \cdots \quad = t_2 t_1 t_2. \tag{19}$$

Combining these identities with the definition of the braiding $\eta_{n,m}$ gives its diagrammatic picture, for instance

$$\eta_{3,2} \equiv \quad \cdots \quad , \qquad \eta_{2,3} \equiv \quad \cdots$$

The next one is almost as easy.

**Lemma 2.4.** *For all $1 \leq i \leq n-1$, $1 \leq j \leq m-1$,*

$$\eta_{n,m} e_i = e_{m+i} \eta_{n,m} \qquad and \qquad \eta_{n,m} e_{n+j} = e_j \eta_{n,m}. \tag{20}$$

*Thus, for all $f \in \mathrm{End}(n)$ and $g \in \mathrm{End}(m)$,*

$$\eta_{n,m}(f \otimes g) = (g \otimes f) \eta_{n,m}. \tag{21}$$

*Proof.* If $1 \leq k \leq i$ and thus $k \leq i < k + n - 1$, the preceding lemma and equation (8) give

$$t_k t_{k+1} \dots t_{k+n-1} e_i = t_k t_{k+1} \dots \underbrace{t_i t_{i+1} e_i}_{e_{i+1} t_i t_{i+1}} t_{i+2} \dots t_{k+n-1} = e_{i+1} t_k t_{k+1} \dots t_{k+n-1}.$$

It follows that

$$\begin{aligned}
\eta_{n,m} e_i &= \prod_{k=1}^{m} (t_k t_{k+1} \dots t_{k+n-1}) e_i \\
&= \prod_{k=2}^{m} (t_k t_{k+1} \dots t_{k+n-1}) \left( e_{i+1} \prod_{k=1}^{1} (t_k t_{k+1} \dots t_{k+n-1}) \right) \\
&= \prod_{k=3}^{m} (t_k t_{k+1} \dots t_{k+n-1}) \left( e_{i+2} \prod_{k=1}^{2} (t_k t_{k+1} \dots t_{k+n-1}) \right) = \dots \\
&= e_{m+i} \eta_{n,m}.
\end{aligned}$$

The second identity in (20) is proved similarly using the second expression of (6). Finally, (21) follows from the fact that $\mathrm{End}(n) \simeq \mathsf{TL}_n$ is generated by the $e_i$. $\qquad\square$

**Lemma 2.5.** *For positive integers $p$ and $n$*

$$\eta_{n,2p}(1_n \otimes z^{\otimes p}) = (z^{\otimes p} \otimes 1_n)\eta_{n,0} \quad and \quad \eta_{0,n}((z^t)^{\otimes p} \otimes 1_n) = (1_n \otimes (z^t)^{\otimes p})\eta_{2p,n} \tag{22}$$

*where $\eta_{0,n} = \eta_{n,0} = 1_n$, $z$ is defined in equation (10), $(z)^t$ is its transpose, and $z^{\otimes p} \equiv \underbrace{z \otimes z \otimes \ldots \otimes z}_{p \text{ times}}$.*

*Proof.* We prove the first identity only as both proofs are nearly identical. We proceed first by induction on $n$ and then on $p$. If $p = n = 1$, the equation is the second of the two equations in (9) that were solved to construct the $t_i$ and obtain (11). Suppose therefore that the result stands for $p = 1$ and some $n \geq 1$. The hexagon identity (4) gives

$$\begin{aligned}
\eta_{n+1,2}(1_{n+1} \otimes z) &= (\eta_{1,2} \otimes 1_n)(1_1 \otimes \eta_{n,2})(1_1 \otimes 1_n \otimes z) \\
&= (\eta_{1,2} \otimes 1_n)(1_1 \otimes (\eta_{n,2}(1_n \otimes z))) \\
&= (\eta_{1,2} \otimes 1_n)(1_1 \otimes z \otimes 1_n) \\
&= (\eta_{1,2}(1_1 \otimes z)) \otimes 1_n \\
&= z \otimes 1_{n+1}.
\end{aligned}$$

Assume then that the result stands for some $p \geq 1$ and all $n \geq 1$. The hexagon identity (3) gives

$$\begin{aligned}
\eta_{n,2p+2}(1_n \otimes z^{\otimes p+1}) &= (1_2 \otimes \eta_{n,2p})(\eta_{n,2} \otimes 1_{2p})(1_n \otimes z \otimes z^{\otimes p}) \\
&= (1_2 \otimes \eta_{n,2p})(\eta_{n,2}(1_n \otimes z) \otimes z^{\otimes p}) \\
&= z \otimes (\eta_{n,2p}(1_n \otimes z^{\otimes p})) \\
&= z^{\otimes p+1} \otimes 1_n
\end{aligned}$$

which ends the proof. $\qquad\square$

In terms of diagrams, this lemma simply states that the two points linked together on the right side of the diagrams in equation (17) can be moved over the underlying links.

$$\eta_{3,2}(z^t \otimes 1_3) = \;\;\;\;\;\;\;\;\;\; = \;\;\;\;\;\;\;\;\;\; = (1_3 \otimes z^t).$$

With these three lemmas, we are now ready to prove the main result of this section.

**Proposition 2.6.** *The category $\widetilde{\mathsf{TL}}$ is braided with a commutor having components*

$$\eta_{r,s} = \prod_{i=1}^{s}\Big(\prod_{j=r-1}^{0} t_{i+j}(r+s)\Big) = \prod_{i=r}^{1}\Big(\prod_{j=0}^{s-1} t_{i+j}(r+s)\Big) \tag{23}$$

*and $t_i(n) = q^{1/2}(1_n + q^{-1}e_i(n))$.*

*Proof.* The category $\widetilde{\mathsf{TL}}$ will be braided if the components $\eta_{r,s}$ are natural isomorphisms satisfying the hexagon axioms. Lemma 2.2 has already showed that the proposed expressions for the components $\eta_{r,s}$ satisfy the hexagon axioms. Moreover, since $t_i(n)$ is invertible, so are the morphisms $\eta_{r,s}$. There remains only the naturality condition to prove. It states the following: For

all pairs $(n, m)$ and $(r, s)$ in Ob $\widetilde{\mathsf{TL}} \times \widetilde{\mathsf{TL}}$ and all pairs of morphisms $(c, d) \in \mathrm{Hom}(n, r) \times \mathrm{Hom}(m, s)$, the following diagram commutes

$$
\begin{array}{ccc}
n \otimes m & \xrightarrow{\;c \otimes d\;} & r \otimes s \\
\eta_{n,m} \downarrow & & \downarrow \eta_{r,s} \\
m \otimes n & \xrightarrow{\;d \otimes c\;} & s \otimes r
\end{array}
$$

Since the Hom-spaces are spanned by diagrams and that the $\eta_{r,s}$ are bilinear, it is sufficient to prove that

$$(d \otimes c)\eta_{n,m} = \eta_{r,s}(c \otimes d) \tag{24}$$

for any $(r, n)$-diagram $c$ and $(s, m)$-diagram $d$.

Consider then $c \in \mathrm{Hom}_{\widetilde{\mathsf{TL}}}(n, r)$ a diagram having $k$ through lines, that is, precisely $k$ nodes on the left side of $c$ are connected to $k$ nodes on its right side. Any such diagram can be expressed as

$$c = a(1_k \otimes z^{\otimes \frac{r-k}{2}})(1_k \otimes (z^t)^{\otimes \frac{n-k}{2}})b, \tag{25}$$

where $a \in \mathrm{End}\, r$ and $b \in \mathrm{End}\, n$. The hexagon identities and lemma 2.5 give

$$
\begin{aligned}
\eta_{r,s}(1_k \otimes z^{\otimes \frac{r-k}{2}} \otimes 1_s) &= (\eta_{k,s} \otimes 1_{r-k})(1_k \otimes \eta_{r-k,s})(1_k \otimes z^{\otimes \frac{r-k}{2}} \otimes 1_s) \\
&= (\eta_{k,s} \otimes 1_{r-k})(1_k \otimes \eta_{r-k,s}(z^{\otimes \frac{r-k}{2}} \otimes 1_s)) \\
&= (\eta_{k,s} \otimes 1_{r-k})(1_k \otimes 1_s \otimes z^{\otimes \frac{r-k}{2}}) \\
&= (1_s \otimes 1_k \otimes z^{\otimes \frac{r-k}{2}})\eta_{k,s}.
\end{aligned}
$$

The same arguments also give

$$\eta_{k,s}(1_k \otimes (z^t)^{\otimes \frac{n-k}{2}} \otimes 1_s) = (1_s \otimes 1_k \otimes (z^t)^{\otimes \frac{n-k}{2}})\eta_{n,s}.$$

Using lemma 2.4, it follows that

$$
\begin{aligned}
\eta_{r,s}(c \otimes 1_s) &= \eta_{r,s}(a \otimes 1_s)(1_k \otimes z^{\otimes \frac{r-k}{2}} \otimes 1_s)(1_k \otimes (z^t)^{\otimes \frac{n-k}{2}} \otimes 1_s)(b \otimes 1_s) \\
&= (1_s \otimes a)\eta_{r,s}(1_k \otimes z^{\otimes \frac{r-k}{2}} \otimes 1_s)(1_k \otimes (z^t)^{\otimes \frac{n-k}{2}} \otimes 1_s)(b \otimes 1_s) \\
&= (1_s \otimes a)(1_s \otimes 1_k \otimes z^{\otimes \frac{r-k}{2}})\eta_{k,s}(1_k \otimes (z^t)^{\otimes \frac{n-k}{2}} \otimes 1_s)(b \otimes 1_s) \\
&= (1_s \otimes a)(1_s \otimes 1_k \otimes z^{\otimes \frac{r-k}{2}})(1_s \otimes 1_k \otimes (z^t)^{\otimes \frac{n-k}{2}})\eta_{n,s}(b \otimes 1_s) \\
&= (1_s \otimes c)\eta_{n,s}.
\end{aligned}
$$

The same steps are used to prove that any diagram $d \in \mathrm{Hom}(m, s)$ with $\ell$ through lines satisfies $\eta_{n,s}(1_n \otimes d) = (d \otimes 1_n)\eta_{n,m}$. Then, for any $(r, n)$-diagram $c$ and $(s, m)$-diagram $d$, these identities give

$$
\begin{aligned}
\eta_{r,s}(c \otimes d) &= \eta_{r,s}(c \otimes 1_s)(1_n \otimes d) = (1_s \otimes c)\eta_{n,s}(1_n \otimes d) \\
&= (1_s \otimes c)(d \otimes 1_n)\eta_{n,m} = (d \otimes c)\eta_{n,m}
\end{aligned}
$$

which closes the proof. $\qquad\square$

Note that with this braiding, $\widetilde{\mathsf{TL}}$ is not symmetric. In general, the element $\eta_{n,m} \circ \eta_{m,n} \in \mathrm{End}(n + m)$ is not even central. For instance, $\eta_{1,2} = q \cdot 1_3 + (e_1 + e_2) + q^{-1}e_2e_1$ and $\eta_{2,1} = q \cdot 1_3 + (e_1 + e_2) + q^{-1}e_1e_2$ and thus

$$\eta_{2,1} \circ \eta_{1,2}e_1 - e_1\eta_{2,1} \circ \eta_{1,2} = q^{-2}(q - q^{-1})(e_1e_2 - e_2e_1) \neq 0.$$

We shall come back to the morphism $\eta_{r,s} \circ \eta_{s,r}$ in 3.3.

## 2.3 The twist $\theta$

The previous section established that the category $\widetilde{\mathsf{TL}}$ is braided. It has even more structure: It has a twist.

A twist $\theta$ on a braided category $\mathcal{C}$ is a natural isomorphism of the identity functor whose components $\{\theta_A \in \mathrm{End}(A), A \in \mathrm{Ob}\,\mathcal{C}\}$ satisfy

$$\theta_{A \otimes B} = \eta_{B,A} \circ \eta_{A,B}(\theta_A \otimes \theta_B), \qquad \text{for all } A \text{ and } B \in \mathrm{Ob}\,\mathcal{C}. \tag{26}$$

This section constructs such a natural isomorphism for the Temperley-Lieb category $\widetilde{\mathsf{TL}}$. The first step toward this goal is to solve a subset of these equations, namely those that have either $r$ or $s$ equal to 1. The next lemma is a corollary of 2.3.

**Lemma 2.7.** *The morphisms $t_i$ satisfy*

$$t_i t_{i+1} \ldots t_{n-1} t_n t_{n-1} \ldots t_{i+1} t_i = t_n t_{n-1} \ldots t_{i+1} t_i t_{i+1} \ldots t_{n-1} t_n. \tag{27}$$

*Proof.* 2.3 provides the cases $t_i t_{i+1} t_i = t_{i+1} t_i t_{i+1}$ for all $i \geq 1$. Then, for a fixed $i$, induction on $n$ gives

$$\begin{aligned}
t_i t_{i+1} \ldots t_{n-1} t_n t_{n-1} \ldots t_{i+1} t_i &= t_i t_{i+1} \ldots [t_{n-1} t_n t_{n-1}] \ldots t_{i+1} t_i \\
&= t_i t_{i+1} \ldots [t_n t_{n-1} t_n] \ldots t_{i+1} t_i \\
&= t_n [t_i t_{i+1} \ldots t_{n-2} t_{n-1} t_{n-2} \ldots t_{i+1} t_i] t_n \\
&= t_n t_{n-1} \ldots t_{i+1} t_i t_{i+1} \ldots t_{n-1} t_n.
\end{aligned}$$

$\square$

The solution of (26) when either $r$ or $s$ is 1 is given by a family of central elements $c_n$ whose main properties are proved in A. (To our knowledge, as an element of $\mathsf{TL}_n$, the element $c_n$ appeared first in Martin's book (see section 6.1 of [13]), but was actually defined much earlier by Chow [12] to study braid groups.)

**Lemma 2.8.** *The central elements $c_n = q^{3n/2}(t_{n-1} \ldots t_2 t_1)^n = q^{3n/2}(t_1 t_2 \ldots t_{n-1})^n$, $n \geq 2$, together with $c_0 = 1_0$ and $c_1 = q^{3/2} 1_1$ satisfy*

$$c_{n+1} = \eta_{1,n} \circ \eta_{n,1}(c_n \otimes c_1) \qquad and \qquad c_{n+1} = \eta_{n,1} \circ \eta_{1,n}(c_1 \otimes c_n), \qquad n \geq 0. \tag{28}$$

*Proof.* Note first that the two equations are trivial for $n = 0$ and, for $n = 1$, they both give

$$\eta_{1,1} \circ \eta_{1,1}(c_1 \otimes c_1) = q^3 (\eta_{1,1})^2 \;\begin{smallmatrix}\bullet\!-\!-\!-\!\bullet\\\bullet\!-\!-\!-\!\bullet\end{smallmatrix}\; = q^3 (\eta_{1,1})^2 = q^3 (t_1)^2 = c_2$$

as claimed. Suppose now that the $c_k$, $k \leq n$, all satisfy both equations. Then

$$\begin{aligned}
q^{-3(n+1)/2} \eta_{1,n} \circ \eta_{n,1}(c_n \otimes c_1) &= (t_n \ldots t_2 t_1)(t_1 t_2 \ldots t_n)(t_{n-1} \ldots t_2 t_1)^n \\
&= (t_n \ldots t_2 t_1)[t_1 t_2 \ldots t_{n-1} t_n t_{n-1} \ldots t_2 t_1](t_{n-1} \ldots t_2 t_1)^{n-1} \\
&= (t_n \ldots t_2 t_1)[(t_n \ldots t_2 t_1)(t_2 t_3 \ldots t_{n-1} t_n)](t_{n-1} \ldots t_2 t_1)^{n-1} \\
&= (t_n \ldots t_2 t_1)^2 (t_2 t_3 \ldots t_{n-1} t_n)(t_{n-1} \ldots t_2 t_1)^{n-1} \\
&= \cdots = (t_n \ldots t_2 t_1)^{n-1} (t_{n-1} t_n)(t_{n-1} \ldots t_2 t_1)^2 \\
&= (t_n \ldots t_2 t_1)^{n-1} [t_{n-1} t_n t_{n-1}](t_{n-2} \ldots t_2 t_1)(t_{n-1} \ldots t_2 t_1)^1 \\
&= (t_n \ldots t_2 t_1)^{n-1} [t_n t_{n-1} t_n](t_{n-2} \ldots t_2 t_1)(t_{n-1} \ldots t_2 t_1)^1 \\
&= (t_n \ldots t_2 t_1)^{n+1} = q^{-3(n+1)/2} c_{n+1}
\end{aligned}$$

where the identity (27) was used repeatedly to transform the sequences of generators between square brackets. Since, by A.2 (d), the central element $c_n$ can be written both as $q^{3n/2}(t_{n-1}\dots t_2 t_1)^n$ and $q^{3n/2}(t_1 t_2 \dots t_{n-1})^n$, a similar argument using the latter form may be used to show that the family $\{c_n\}$ also solves the second identity in (28). $\qquad\square$

The diagram representing $c_n$ is quite convoluted, but we nevertheless illustrate the first identity in (28) for $n = 3$ and with the symbol $\sim$ meaning "equal up to a power of $q$":

$$c_4 \sim \quad\sim\quad \sim (\eta_{1,3}\eta_{3,1})(c_3 \otimes c_1). \tag{29}$$

The final identification of the twist $\{\theta_n, n \in \mathbb{N}_0\}$ requires yet another technical lemma.

**Lemma 2.9.** *The commutor $\{\eta_{r,s}\}$ satisfies*

$$\eta_{s+1,r-1}(\eta_{s,1} \otimes 1_{r-1}) = \eta_{s,r}(1_s \otimes \eta_{1,r-1}), \tag{30}$$
$$\eta_{r-1,s+1}(1_{r-1} \otimes \eta_{1,s}) = \eta_{r,s}(\eta_{r-1,1} \otimes 1_s). \tag{31}$$

*for all $r$ and $s \in \mathbb{N}_0$ such that all indices in these identities are non-negative.*

*Proof.* The proof of the first identity proceeds as follows and uses the explicit form (23) of the commutor:

$$\eta_{s+1,r-1}(\eta_{s,1} \otimes 1_{r-1}) = (1_{r-1} \otimes \eta_{s,1})\eta_{s+1,r-1} \qquad \text{by the naturality (24) of } \eta$$

$$= (1_{r-1} \otimes (t_1 t_2 \dots t_s)) \cdot \prod_{i=s+1}^{1} \prod_{j=0}^{r-2} t_{i+j}$$

$$= (t_r t_{r+1} \dots t_{r+s-1})[(t_{r-1} \dots t_2 t_1)(t_r \dots t_3 t_2)\dots(t_{r+s-2}\dots t_{s+1}t_s)](t_{r+s-1}\dots t_{s+2}t_{s+1})$$

and then moving the leftmost $t_r, t_{r+1},\dots$, and $t_{r+s-1}$ to their rightmost possible positions within the square brackets

$$= [(t_r t_{r-1} \dots t_2 t_1)(t_{r+1} t_r \dots t_3 t_2)\dots(t_{r+s-1} t_{r+s-2} \dots t_{s+1} t_s)](t_{r+s-1}\dots t_{s+2}t_{s+1})$$

$$= \prod_{i=s}^{1} \prod_{j=0}^{r-1} t_{i+j} \cdot (1_s \otimes \eta_{1,r-1}) = \eta_{s,r}(1_s \otimes \eta_{1,r-1}).$$

The second identity is obtained from the first by the substitutions $r - 1 \to s, s + 1 \to r$. $\qquad\square$

In terms of diagrams, the identity (30) (with $s = r = 3$) is

$$\eta_{4,2}(\eta_{3,1} \otimes 1_2) = \quad = \quad = \eta_{3,3}(1_3 \otimes \eta_{1,2}). \tag{32}$$

This lemma simplifies considerably the proof of the existence of the twist.

**Proposition 2.10.** *The morphisms in* $\mathrm{End}(n)$ *given by the multiplication by* $\theta_n = c_n, n \in \mathbb{N}_0$, *define a natural isomorphism $\theta$ between the identity functor and itself, and is a twist for the commutor $\eta_{\cdot}$.*

*Proof.* [3] The first step is to prove that the family $\{\theta_n = c_n, n \in \mathbb{N}_0\}$ satisfy (26), which amounts to establish

$$\eta_{s+1,r-1} \circ \eta_{r-1,s+1}(\theta_{r-1} \otimes \theta_{s+1}) = \eta_{s,r}\eta_{r,s}(\theta_r \otimes \theta_s). \tag{33}$$

Indeed 2.8 has proved $\eta_{1,n}\eta_{n,1}(\theta_n \otimes \theta_1) = \theta_{n+1}$ and, starting from the latter, the above identity proves recursively the identities (26) for all $r$ and $s$ such that $n+1 = r+s$. Note that equation (28) holds also for the components $\theta_n = c_n$.

We first rewrite $(\theta_{r-1} \otimes \theta_{s+1})$ in terms of $(\theta_r \otimes \theta_s)$:

$$
\begin{aligned}
(\theta_{r-1} \otimes \theta_{s+1}) &= (\theta_{r-1} \otimes 1_{s+1})(1_{r-1} \otimes \theta_{s+1}) \\
&= (\theta_{r-1} \otimes 1_{s+1})(1_{r-1} \otimes (\eta_{s,1}\eta_{1,s}(\theta_1 \otimes \theta_s))), \qquad \text{by (28)}, \\
&= (\theta_{r-1} \otimes 1_{s+1})(1_{r-1} \otimes \eta_{s,1}\eta_{1,s})(1_{r-1} \otimes \theta_1 \otimes \theta_s) \\
&= (1_{r-1} \otimes \eta_{s,1}\eta_{1,s})(\theta_{r-1} \otimes \theta_1 \otimes 1_s)(1_r \otimes \theta_s), \qquad \text{because } \theta_1 = 1_1, \\
&= (1_{r-1} \otimes \eta_{s,1}\eta_{1,s})(\eta_{r-1,1}^{-1}\eta_{1,r-1}^{-1}\theta_r \otimes 1_s)(1_r \otimes \theta_s), \qquad \text{again by (28)}, \\
&= (1_{r-1} \otimes \eta_{s,1}\eta_{1,s})(\eta_{r-1,1}^{-1}\eta_{1,r-1}^{-1} \otimes 1_s)(\theta_r \otimes \theta_s).
\end{aligned}
$$

It is then sufficient to prove

$$\eta_{s+1,r-1}\eta_{r-1,s+1}(1_{r-1} \otimes \eta_{s,1}\eta_{1,s})(\eta_{r-1,1}^{-1}\eta_{1,r-1}^{-1} \otimes 1_s) = \eta_{s,r}\eta_{r,s}.$$

With the technical 2.9, this is now straightforward:

$$
\begin{aligned}
&\eta_{s+1,r-1}\eta_{r-1,s+1}(1_{r-1} \otimes \eta_{s,1}\eta_{1,s})(\eta_{r-1,1}^{-1}\eta_{1,r-1}^{-1} \otimes 1_s) \\
&= \eta_{s+1,r-1}[\eta_{r-1,s+1}(1_{r-1} \otimes \eta_{s,1})](1_{r-1} \otimes \eta_{1,s})(\eta_{r-1,1}^{-1}\eta_{1,r-1}^{-1} \otimes 1_s) \\
&= [\eta_{s+1,r-1}(\eta_{s,1} \otimes 1_{r-1})][\eta_{r-1,s+1}(1_{r-1} \otimes \eta_{1,s})(\eta_{r-1,1}^{-1} \otimes 1_s)](\eta_{1,r-1}^{-1} \otimes 1_s),
\end{aligned}
$$

by the naturality (24) of $\eta$,

$$
\begin{aligned}
&= \eta_{s,r}(1_s \otimes \eta_{1,r-1})\eta_{r,s}(\eta_{1,r-1}^{-1} \otimes 1_s), \qquad \text{by (30) and (31)}, \\
&= \eta_{s,r}(1_s \otimes \eta_{1,r-1})(1_s \otimes \eta_{1,r-1}^{-1})\eta_{r,s}, \qquad \text{again by naturality}, \\
&= \eta_{s,r}\eta_{r,s},
\end{aligned}
$$

which ends the proof of (26).

It remains to prove that $\theta$ defines a natural isomorphism of the identity functor, i.e. that for all $f \in \text{Hom}(m,n)$

$$\theta_n \circ f = f \circ \theta_m. \tag{34}$$

Let $f \in \text{Hom}(m,n)$ be a diagram with $k$ through lines. As before (see equation (25) of the proof of proposition 2.6), such a diagram can be written as

$$a \circ (1_k \otimes z^{\otimes(n-k)/2}) \circ (1_k \otimes \bar{z}^{\otimes(m-k)/2}) \circ b, \tag{35}$$

for some $a \in \text{End}(n)$, $b \in \text{End}(m)$. Since $\theta_n$ is central in $\text{End}(n)$ by proposition A.2,

$$
\begin{aligned}
\theta_n \circ f &\overset{1}{=} a \circ \theta_n \circ (1_k \otimes z^{\otimes(n-k)/2}) \circ (1_k \otimes \bar{z}^{\otimes(m-k)/2}) \circ b \\
&\overset{2}{=} a \circ \eta_{n-k,k}\eta_{k,n-k}(\theta_k \otimes \theta_{n-k}z^{\otimes(n-k)/2}) \circ (1_k \otimes \bar{z}^{\otimes(m-k)/2}) \circ b \\
&\overset{3}{=} a \circ (1_k \otimes z^{\otimes(n-k)/2}) \circ \theta_k \circ (1_k \otimes \bar{z}^{\otimes(m-k)/2}) \circ b \\
&\overset{4}{=} a \circ (1_k \otimes z^{\otimes(n-k)/2}) \circ (1_k \otimes \bar{z}^{\otimes(m-k)/2})\eta_{m-k,k}\eta_{k,m-k}(\theta_k \otimes \theta_{m-k}) \circ b \\
&\overset{5}{=} f \circ \theta_m.
\end{aligned}
$$

---

[3]The very last line of this proof rests on a basic property of standard modules $\mathsf{S}_{n,k}$ over $\mathsf{TL}_n$. The reader not familiar with these might want to postpone the reading of the proof after the introduction of these modules in the next section and the computation of $\gamma_{n,k}$ in part (b) of proposition A.2.

Steps 2 and 5 were obtained by using equation (26). Steps 3 and 4 rest on two observations. (It is here that the factor $q^{3n/2}$ plays an essential role!) First $\eta$ is a natural transformation and thus $\eta_{n-k,k}\eta_{k,n-k}(1_k \otimes \theta_{n-k}z^{\otimes(n-k)/2}) = (1_k \otimes \theta_{n-k}z^{\otimes(n-k)/2})\eta_{0,k}\eta_{k,0} = (1_k \otimes \theta_{n-k}z^{\otimes(n-k)/2})$. Second proposition A.2 gives $\theta_{n-k}z^{\otimes(n-k)/2} = z^{\otimes(n-k)/2}$, since $\text{Hom}(n,0) \simeq S_{n,0}$ as left $\text{End}(n)$-modules. □

## 3  The category of modules over Temperley-Lieb algebras

### 3.1  Braiding modules

The representation theory of the family of $\mathsf{TL}_n, n \in \mathbb{N}_0$, was cast into a categorical framework by Graham and Lehrer [1] as follows. Let $\mathsf{F} \in \text{Funct}(\widetilde{\mathsf{TL}}, \text{Vect}_\mathbb{C})$ be a functor from the category $\widetilde{\mathsf{TL}}$ to that of finite-dimensional vector spaces over $\mathbb{C}$. Then each $\mathsf{F}(n), n \in \mathbb{N}_0$, is a vector space and each $\mathsf{F}(\alpha)$, for $\alpha \in \text{Hom}(n,n) \simeq \mathsf{TL}_n$ is a linear map $\mathsf{F}(n) \to \mathsf{F}(n)$. Since $\mathsf{F}$ preserves composition, $\mathsf{F}(n)$ is naturally a $\mathsf{TL}_n$-module, but the functor $\mathsf{F}$ is somewhat richer than a choice of a $\mathsf{TL}_n$-module for each $n \geq 0$. Indeed the functor $\mathsf{F}$ also gives linear maps $\mathsf{F}(\gamma) : \mathsf{F}(n) \to \mathsf{F}(m)$ for all $\gamma \in \text{Hom}(n,m)$ between modules over distinct algebras of the Temperley-Lieb family. These linear maps must also preserve the composition of diagrams. We now give examples of such functors taken from [1].

Let $k \in \mathbb{N}_0$. A functor $\mathsf{S}_k \in \text{Funct}(\widetilde{\mathsf{TL}}, \text{Vect}_\mathbb{C})$ is defined as follows. If the parities of $n$ and $k$ are distinct, then the vector space $\mathsf{S}_k(n)$ is set to 0. If their parities coincide, then $\mathsf{S}_k(n)$ is the formal span of $(n,k)$-diagrams with exactly $k$ through lines. If $\alpha \in \text{Hom}(n,m)$ with $m,n$ and $k$ having the same parity, then $\mathsf{S}_k(\alpha) : \mathsf{S}_k(n) \to \mathsf{S}_k(m)$ is the linear map defined by its action on $(n,k)$-diagrams with $k$ through lines. If $\gamma \in \mathsf{S}_k(n)$ is such diagram, then $\mathsf{S}_k(\alpha)\gamma \in \text{Hom}(k,m)$ is $\alpha \circ \gamma$ if $\alpha \circ \gamma$ has $k$ through lines and 0 otherwise. For all other $\alpha \in \text{Hom}(m',n')$, that is with $m'$ or $n'$ not sharing the parity of $k$, the linear map $\mathsf{S}_k(\alpha)$ is zero. It is straightforward to check that $\mathsf{S}_k$ is a functor and that $\mathsf{S}_k(n)$ is the usual *standard* or *cellular* $\mathsf{TL}_n$-modules $\mathsf{S}_{n,k}$. The functor $\mathsf{S}_k$ just described is simply $\mathsf{S}_k(-) = \text{Hom}(k,-) \otimes_{\mathsf{TL}_k} \mathsf{S}_{k,k}$ where $\mathsf{S}_{k,k}$ is the one-dimensional standard $\mathsf{TL}_k$-module.

Any module $\mathsf{M}$ over $\mathsf{TL}_m$ is the evaluation of a certain functor $\mathsf{F}$ at $m$, for example the functor $\text{Hom}(m,-) \otimes_{\mathsf{TL}_m} \mathsf{M}$, that will be denoted by either $\mathsf{F}_{m,\mathsf{M}}$ or simply $\mathsf{F}_\mathsf{M}$. (Recall that $\text{Hom}(m,m) = \mathsf{TL}_m$ and $\mathsf{F}_\mathsf{M}(m) = \text{Hom}(m,m) \otimes_{\mathsf{TL}_m} \mathsf{M} \simeq \mathsf{M}$.) We shall use the letter $\mathsf{I} \in \text{Funct}(\widetilde{\mathsf{TL}}, \text{Vect}_\mathbb{C})$ for the functor $\mathsf{I}(-) = \text{Hom}_{\widetilde{\mathsf{TL}}}(0,-) \otimes_{\mathsf{TL}_0} \mathsf{TL}_0 \simeq \text{Hom}_{\widetilde{\mathsf{TL}}}(0,-)$. Recall that $\mathsf{TL}_0 \simeq \mathbb{C}$ and $\text{Hom}(0,0) = \mathbb{C}$. The functor $\mathsf{I}$ is thus $\mathsf{F}_{0,\mathsf{TL}_0}$.

Our first step is to define a category of modules associated to $\widetilde{\mathsf{TL}}$ compatible with Graham and Lehrer's framework. The examples given above, $\mathsf{F}_{m,\mathsf{M}}$ and $\mathsf{F}_{k,\mathsf{S}_{k,k}}$, should be objects of this category, but slightly more general functors will be useful. Let $m$ be a positive integer and $\overline{m} = \{m_1, m_2, \ldots, m_a\}$, where the $m_i \geq 1, 1 \leq i \leq a$, be a partition of $m = \sum_i m_i$. For each $i$, let $\mathsf{M}_i$ be a $\mathsf{TL}_{m_i}$-module. Clearly $\overline{\mathsf{M}} = \mathsf{M}_1 \otimes_\mathbb{C} \mathsf{M}_2 \otimes_\mathbb{C} \cdots \otimes_\mathbb{C} \mathsf{M}_a$ is a module over the product $\mathsf{TL}_{\overline{m}} \equiv \mathsf{TL}_{m_1} \otimes_{\widetilde{\mathsf{TL}}} \mathsf{TL}_{m_2} \otimes_{\widetilde{\mathsf{TL}}} \cdots \otimes_{\widetilde{\mathsf{TL}}} \mathsf{TL}_{m_a}$ of the algebras $\mathsf{TL}_{m_i}$. The data $(\overline{m}, \overline{\mathsf{M}})$ define naturally a functor $\mathsf{F}_{\overline{m},\overline{\mathsf{M}}} \in \text{Funct}(\widetilde{\mathsf{TL}}, \text{Vect}_\mathbb{C})$ (or simply $\mathsf{F}_{\overline{\mathsf{M}}}$) by

$$\mathsf{F}_{\overline{m},\overline{\mathsf{M}}}(-) = \text{Hom}_{\widetilde{\mathsf{TL}}}(m,-) \otimes_{\mathsf{TL}_{\overline{m}}} \overline{\mathsf{M}} \tag{36}$$
$$= \text{Hom}_{\widetilde{\mathsf{TL}}}(m,-) \otimes_{\mathsf{TL}_{m_1} \otimes_{\widetilde{\mathsf{TL}}} \mathsf{TL}_{m_2} \otimes_{\widetilde{\mathsf{TL}}} \cdots \otimes_{\widetilde{\mathsf{TL}}} \mathsf{TL}_{m_a}} (\mathsf{M}_1 \otimes_\mathbb{C} \mathsf{M}_2 \otimes_\mathbb{C} \cdots \otimes_\mathbb{C} \mathsf{M}_a),$$

with the action on morphisms $f : n \to k$ given by

$$\mathsf{F}_{\overline{m},\overline{\mathsf{M}}}(f)(\alpha \otimes_{\mathsf{TL}_{\overline{m}}} x) \equiv (f \circ \alpha) \otimes_{\mathsf{TL}_{\overline{m}}} x, \tag{37}$$

for $\alpha \in \mathrm{Hom}(m,n)$ and $x \in \overline{\mathsf{M}}$.

Note that, given a pair $(\overline{m}, \overline{\mathsf{M}})$, the functor $\mathsf{F}_{\overline{m},\overline{\mathsf{M}}}$ is isomorphic to a functor $\mathsf{F}_{m,\mathsf{M}'}$ for a certain $\mathsf{TL}_m$-module $\mathsf{M}'$. Indeed

$$\mathsf{F}_{\overline{m},\overline{\mathsf{M}}}(-) = \mathrm{Hom}_{\widetilde{\mathsf{TL}}}(m,-) \otimes_{\mathsf{TL}_{\overline{m}}} \overline{\mathsf{M}} \simeq \mathrm{Hom}_{\widetilde{\mathsf{TL}}}(m,-) \otimes_{\mathsf{TL}_m} (\mathsf{TL}_m \otimes_{\mathsf{TL}_{\overline{m}}} \overline{\mathsf{M}}) = \mathsf{F}_{m,\overline{\mathsf{M}}\uparrow}(-)$$

where $\overline{\mathsf{M}}\uparrow \equiv \mathsf{M}'$ is the induced module from $\mathsf{TL}_{m_1} \otimes_{\widetilde{\mathsf{TL}}} \mathsf{TL}_{m_2} \otimes_{\widetilde{\mathsf{TL}}} \cdots \otimes_{\widetilde{\mathsf{TL}}} \mathsf{TL}_{m_a}$ to $\mathsf{TL}_m$. Despite this observation, the definition (36) will show its usefulness below.

**Definition 2.** *The category* $\mathrm{Mod}_{\widetilde{\mathsf{TL}}}$ *has as objects the functors* $\mathsf{F}_{\overline{m},\overline{\mathsf{M}}}$ *for all partitions* $\overline{m}$ *and choices of modules* $\overline{\mathsf{M}}$ *(together with the functor* $\mathsf{I}(-) = \mathrm{Hom}(0,-) \otimes_{\mathsf{TL}_0} \mathsf{TL}_0$*) with their direct sums (as functors), and as morphisms* $\mathrm{Hom}_{\mathrm{Mod}_{\widetilde{\mathsf{TL}}}}(\mathsf{F}_{\overline{m},\overline{\mathsf{M}}}, \mathsf{F}_{\overline{n},\overline{\mathsf{N}}})$ *the natural transformations* $\mathrm{Nat}(\mathsf{F}_{\overline{m},\overline{\mathsf{M}}}, \mathsf{F}_{\overline{n},\overline{\mathsf{N}}})$ *between these functors.*

Note that, since clearly $\mathrm{Mod}_{\widetilde{\mathsf{TL}}} \subset \mathrm{Funct}(\widetilde{\mathsf{TL}}, \mathrm{Vect}_{\mathbb{C}})$, the direct sums are defined in the same way on both. In particular, $\mathsf{F}_{n,\mathsf{N}_1 \oplus \mathsf{N}_2} \simeq \mathsf{F}_{n,\mathsf{N}_1} \oplus \mathsf{F}_{n,\mathsf{N}_2}$.

Here is a simple example of a natural transformation between $\mathsf{F}_\mathsf{M}$ and $\mathsf{F}_\mathsf{N}$ where both $\mathsf{M}$ and $\mathsf{N}$ are $\mathsf{TL}_m$-modules with $m = n$. In this case both partitions $\overline{m}$ and $\overline{n}$ are simply $\{m\}$. Let $f : \mathsf{M} \to \mathsf{N}$ be a morphism. Define $\phi_f : \mathsf{F}_\mathsf{M} \to \mathsf{F}_\mathsf{N}$ by the linear transformations $\phi_f(k) : \mathsf{F}_\mathsf{M}(k) \to \mathsf{F}_\mathsf{N}(k)$

$$a \otimes_{\mathsf{TL}_m} x \longmapsto a \otimes_{\mathsf{TL}_m} f x$$

if $a \in \mathrm{Hom}_{\widetilde{\mathsf{TL}}}(m,k)$ and $x \in \mathsf{M}$. Clearly this is well-defined: $f(a \otimes_{\mathsf{TL}_m} cx) = a \otimes_{\mathsf{TL}_m} f(cx) = ac \otimes_{\mathsf{TL}_m} f(x) = f(ac \otimes_{\mathsf{TL}_m} x)$ for any $c \in \mathsf{TL}_m$. The naturality of $\phi_f$ is easily checked. For $b \in \mathrm{Hom}(k,l)$

$$\begin{aligned}
\mathsf{F}_\mathsf{N}(b) \circ \phi_f(k)(a \otimes_{\mathsf{TL}_m} x) &= \mathsf{F}_\mathsf{N}(b)(a \otimes_{\mathsf{TL}_m} f x) \\
&= (ba) \otimes_{\mathsf{TL}_m} f x \\
&= \phi_f(l) \circ \mathsf{F}_\mathsf{M}(b)(a \otimes_{\mathsf{TL}_m} x),
\end{aligned}$$

that is $\mathsf{F}_\mathsf{N}(b) \circ \phi_f(k) = \phi_f(l) \circ \mathsf{F}_\mathsf{M}(b)$ if $b \in \mathrm{Hom}(k,l)$. Other examples are given below.

Let $\mathsf{M}$ and $\mathsf{N}$ be a $\mathsf{TL}_m$- and a $\mathsf{TL}_n$-module, respectively. A fusion product $\mathsf{M} \times_f \mathsf{N}$ was first defined by Read and Saleur [2] and later on computed systematically by Gainutdinov and Vasseur [3] and Belletête [4]. To endow $\mathrm{Mod}_{\widetilde{\mathsf{TL}}}$ with a braided structure,[4] we extend their definition to pairs $(\overline{m}, \overline{\mathsf{M}})$ and $(\overline{n}, \overline{\mathsf{N}})$ of partitions and choices of modules:

$$\overline{\mathsf{M}} \times_f \overline{\mathsf{N}} \equiv \mathsf{TL}_{m+n} \otimes_{\mathsf{TL}_{\overline{m}} \otimes_{\widetilde{\mathsf{TL}}} \mathsf{TL}_{\overline{n}}} (\overline{\mathsf{M}} \otimes_{\mathbb{C}} \overline{\mathsf{N}}). \tag{38}$$

The fusion $\overline{\mathsf{M}} \times_f \overline{\mathsf{N}}$ is thus a left $\mathsf{TL}_{m+n}$-module. This (slightly more general) definition makes the introduction of a bifunctor $-_1 \times_f -_2$ on $\mathrm{Mod}_{\widetilde{\mathsf{TL}}}$ straightforward:

$$\mathsf{F}_{\overline{m},\overline{\mathsf{M}}} \times_f \mathsf{F}_{\overline{n},\overline{\mathsf{N}}}(-) = \mathrm{Hom}(m+n,-) \otimes_{\mathsf{TL}_{\overline{m}} \otimes_{\widetilde{\mathsf{TL}}} \mathsf{TL}_{\overline{n}}} (\overline{\mathsf{M}} \otimes_{\mathbb{C}} \overline{\mathsf{N}}) \tag{39}$$

which can be rewritten as

$$\begin{aligned}
&\simeq \mathrm{Hom}(m+n,-) \otimes_{\mathsf{TL}_{m+n}} (\mathsf{TL}_{m+n} \otimes_{\mathsf{TL}_{\overline{m}} \otimes_{\widetilde{\mathsf{TL}}} \mathsf{TL}_{\overline{n}}} (\overline{\mathsf{M}} \otimes_{\mathbb{C}} \overline{\mathsf{N}})) \\
&\simeq \mathrm{Hom}(m+n,-) \otimes_{\mathsf{TL}_{m+n}} (\overline{\mathsf{M}} \times_f \overline{\mathsf{N}}) \\
&\simeq \mathsf{F}_{m+n, \overline{\mathsf{M}} \times_f \overline{\mathsf{N}}}(-). \tag{40}
\end{aligned}$$

---

[4]The concept of *fusion category* exists in the literature (see, for example [10]). Even though it describes categories equipped with a bifunctor $-_1 \otimes -_2$ (among other structures), the categories of modules under study here are not fusion categories, as the latter contain only semisimple modules.

The bifunctor's action on morphism is defined in the obvious way, that is, for $g : F_{m,M} \to F_{u,U}$ a morphism, then the morphism $g \times_f F_{n,N}$ is defined through its components (removing the overhead bars for simplicity)

$$(g \times_f F_{n,N})_k : \mathrm{Hom}(m+n,k) \otimes_{TL_m \otimes TL_n} (M \otimes_{\mathbb{C}} N) \to \mathrm{Hom}(u+n,k) \otimes_{TL_u \otimes TL_n} (U \otimes_{\mathbb{C}} N)$$
$$\alpha \otimes_{TL_m \otimes TL_n} (x \otimes_{\mathbb{C}} y) \to \sum_{z \in U} \alpha \circ (\gamma_{z,x} \otimes_{\widetilde{TL}} 1_n) \otimes_{TL_m \otimes TL_n} (z \otimes_{\mathbb{C}} y),$$
$$(41)$$

for all $\alpha \in \mathrm{Hom}(m+n,k)$, $x \in M$, $y \in N$, and $\{\gamma_{z,x}\}_{z \in U} \subset \mathrm{Hom}(u,m)$ is such that $g_m(1_m \otimes_{TL_m} x) = \sum_{z \in U} \gamma_{z,x} \otimes_{TL_u} z$. The morphism $F_{m,M} \times_f h$ for $h : F_{n,N} \to F_{u,U}$ is then defined in an analogous manner.

The functor $I(-) = \mathrm{Hom}(0,-)$ acts as the identity for this product:

$$I \times_f F_{\overline{m},\overline{M}}(-) = \mathrm{Hom}(0+m,-) \otimes_{TL_0 \otimes_{\widetilde{TL}} TL_{\overline{m}}} (TL_0 \otimes_{\mathbb{C}} \overline{M})$$
$$\simeq \mathrm{Hom}(m,-) \otimes_{TL_{\overline{m}}} \overline{M}, \qquad \text{since } TL_0 \simeq \mathbb{C}$$
$$= F_{\overline{m},\overline{M}}(-)$$

and this isomorphism $I \times_f F_{\overline{m},\overline{M}} \mapsto F_{\overline{m},\overline{M}}$ defines a left unitor $\lambda$ on $\mathrm{Mod}_{\widetilde{TL}}$. A right one $\rho$ is defined similarly.

With the definition of the bifunctor $\times_f$, more examples can be given of functorial morphisms between objects of $\mathrm{Mod}_{\widetilde{TL}}$. The following ones turn out to be natural isomorphisms.

**Proposition 3.1.** *If $\beta \neq 0$, then $F_{2,S_{2,0}} \simeq F_{0,TL_0}$.*

*Proof.* Let $\varphi : F_{2,S_{2,0}} \to F_{0,TL_0}$ be defined by

$$\varphi(k) : F_{2,S_{2,0}}(k) \simeq F_{0,TL_0}(k)$$
$$f \otimes_{TL_2} \text{ⅇ} \mapsto f \circ \text{ⅇ} \otimes_{\mathbb{C}} (1/\beta),$$

where $f \in \mathrm{Hom}_{\widetilde{TL}}(2,k)$. Then, if $a \in \mathrm{Hom}(k,l)$:

$$\varphi(l)(a \circ f \otimes_{TL_2} \text{ⅇ}) = a \circ f \circ \text{ⅇ} \otimes_{\mathbb{C}} (1/\beta) = a \circ (\varphi(k)(f \otimes_{TL_2} \text{ⅇ})),$$

that is, $\varphi$ is a morphism or, in other words $\varphi \in \mathrm{Nat}(F_{2,S_{2,0}}, F_{0,TL_0})$ Clearly $\varphi$ has an inverse defined by $\varphi^{-1}(k)(f \otimes_{\mathbb{C}} c) = c(f \circ \text{ⅎ}) \otimes_{TL_2} \text{ⅇ}$ for $c \in \mathbb{C} = TL_0$ and $f \in \mathrm{Hom}(0,k)$, and $\varphi^{-1}$ is also a natural transformation. Thus $\varphi$ is a natural isomorphism. $\qquad \square$

**Corollary 3.2.** *Let $\beta \neq 0$ and $F_{\overline{m},\overline{M}} \in \mathrm{Ob}(\mathrm{Mod}_{\widetilde{TL}})$. Then $F_{\overline{m},\overline{M}} \simeq F_{m+2,\overline{M} \times_f S_{2,0}}$. In particular*

$$F_{k,S_{k,k}} \simeq F_{k+2n,S_{k+2n,k}}, \qquad \text{for all } k,n \geq 0. \tag{42}$$

*Proof.* By equation (40),

$$F_{m+2,\overline{M} \times_f S_{2,0}} \simeq F_{\overline{m},\overline{M}} \times_f F_{2,S_{2,0}} \simeq F_{\overline{m},\overline{M}} \times_f F_{0,TL_0} \simeq F_{\overline{m},\overline{M}} \times_f I \simeq F_{\overline{m},\overline{M}}.$$

The isomorphism (42) follows from the identity $S_{k+2i,k} \times_f S_{2,0} \simeq S_{k+2(i+1),k}$ that holds when $\beta \neq 0$ (see Prop. A.1 of [4]). $\qquad \square$

The set of natural transformations from the functor $F_{n,TL_n}$, where $TL_n$ is seen here as the left regular module, to the functor $F_{\overline{m},\overline{M}}$ can be made explicit. Indeed, by definition of $F_{n,TL_n}(-) = \mathrm{Hom}(n,-) \otimes_{TL_n} TL_n \simeq \mathrm{Hom}(n,-)$. Thus

$$\mathrm{Nat}(F_{n,TL_n}, F_{\overline{m},\overline{M}}) \simeq \mathrm{Nat}(\mathrm{Hom}(n,-), F_{\overline{m},\overline{M}}) \simeq F_{\overline{m},\overline{M}}(n), \qquad \text{as vector spaces,}$$

where the last isomorphism follows from Yoneda's lemma. In particular

$$\mathrm{Hom}_{\mathrm{Mod}_{\widetilde{\mathsf{TL}}}}(\mathsf{F}_{0,\mathbb{C}}, \mathsf{F}_{\overline{m},\overline{M}}) = \mathrm{Nat}(\mathsf{F}_{0,\mathbb{C}}, \mathsf{F}_{\overline{m},\overline{M}}) \simeq \mathsf{F}_{\overline{m},\overline{M}}(0) \simeq \mathsf{S}_{m,0} \otimes_{\mathsf{TL}_{\overline{m}}} \overline{M},$$

again as vector spaces.

The definition (36) of objects in $\mathrm{Mod}_{\widetilde{\mathsf{TL}}}$ allows for an easy definition of an associator, also noted $\alpha$, on $\mathrm{Mod}_{\widetilde{\mathsf{TL}}}$. Let $\mathsf{F}_{\overline{l},\overline{L}}$, $\mathsf{F}_{\overline{m},\overline{M}}$ and $\mathsf{F}_{\overline{n},\overline{N}}$ be three objects in $\mathrm{Mod}_{\widetilde{\mathsf{TL}}}$. Then

$$\mathsf{F}_{\overline{l},\overline{L}} \times_f (\mathsf{F}_{\overline{m},\overline{M}} \times_f \mathsf{F}_{\overline{n},\overline{N}}) = \mathrm{Hom}(l+m+n,-) \otimes_{\mathsf{TL}_{\overline{l}} \otimes_{\widetilde{\mathsf{TL}}} \mathsf{TL}_{m+n}} (\overline{L} \otimes_{\mathbb{C}} (\mathsf{TL}_{m+n} \otimes_{\mathsf{TL}_{\overline{m}} \otimes_{\widetilde{\mathsf{TL}}} \mathsf{TL}_{\overline{n}}} (\overline{M} \otimes_{\mathbb{C}} \overline{N})))$$

$$\simeq \mathrm{Hom}(l+m+n,-) \otimes_{\mathsf{TL}_{\overline{l}} \otimes_{\widetilde{\mathsf{TL}}} \mathsf{TL}_{m+n}} [(\mathsf{TL}_{\overline{l}} \otimes_{\widetilde{\mathsf{TL}}} \mathsf{TL}_{m+n}) \otimes_{\mathsf{TL}_{\overline{l}} \otimes_{\widetilde{\mathsf{TL}}} \mathsf{TL}_{\overline{m}} \otimes_{\widetilde{\mathsf{TL}}} \mathsf{TL}_{\overline{n}}} (\overline{L} \otimes_{\mathbb{C}} (\overline{M} \otimes_{\mathbb{C}} \overline{N}))]$$

$$\simeq \mathrm{Hom}(l+m+n,-) \otimes_{\mathsf{TL}_{\overline{l}} \otimes_{\widetilde{\mathsf{TL}}} \mathsf{TL}_{\overline{m}} \otimes_{\widetilde{\mathsf{TL}}} \mathsf{TL}_{\overline{n}}} (\overline{L} \otimes_{\mathbb{C}} (\overline{M} \otimes_{\mathbb{C}} \overline{N})).$$

Similarly

$$(\mathsf{F}_{\overline{l},\overline{L}} \times_f \mathsf{F}_{\overline{m},\overline{M}}) \times_f \mathsf{F}_{\overline{n},\overline{N}} \simeq \mathrm{Hom}(l+m+n,-) \otimes_{\mathsf{TL}_{\overline{l}} \otimes_{\widetilde{\mathsf{TL}}} \mathsf{TL}_{\overline{m}} \otimes_{\widetilde{\mathsf{TL}}} \mathsf{TL}_{\overline{n}}} ((\overline{L} \otimes_{\mathbb{C}} \overline{M}) \otimes_{\mathbb{C}} \overline{N}).$$

Let $k \in \mathbb{N}_0$, $f \in \mathrm{Hom}(l+m+n,k)$ and $x \in \overline{L}, y \in \overline{M}, z \in \overline{N}$. Then the associator $\alpha_{(\overline{l},\overline{L}),(\overline{m},\overline{M}),(\overline{n},\overline{N})}$ must act as

$$f \otimes_{\mathsf{TL}_{\overline{l}} \otimes_{\widetilde{\mathsf{TL}}} \mathsf{TL}_{\overline{m}} \otimes_{\widetilde{\mathsf{TL}}} \mathsf{TL}_{\overline{n}}} ((x \otimes_{\mathbb{C}} y) \otimes_{\mathbb{C}} z) \longmapsto f \otimes_{\mathsf{TL}_{\overline{l}} \otimes_{\widetilde{\mathsf{TL}}} \mathsf{TL}_{\overline{m}} \otimes_{\widetilde{\mathsf{TL}}} \mathsf{TL}_{\overline{n}}} (x \otimes_{\mathbb{C}} (y \otimes_{\mathbb{C}} z)).$$

The verification of the triangle and pentagon axioms for these unitors $\lambda, \rho$ and associator $\alpha$ then mimics that for the usual tensor product of vector spaces.

A lighter notation will be used from now on. The tensor signs $\otimes_{\mathbb{C}}$ and $\otimes_{\mathsf{TL}_{\overline{m}} \otimes_{\widetilde{\mathsf{TL}}} \mathsf{TL}_{\overline{n}}}$ will be replaced by $\otimes$ and $\otimes_{\overline{m},\overline{n}}$ respectively and the functor $\mathsf{F}_{\overline{m},\overline{M}}$ by $\mathsf{F}_{\overline{M}}$. Furthermore, when appearing in indices, we shall write only $\overline{M}$ instead of $\mathsf{F}_{\overline{M}}$. The braiding $\eta$ of $\widetilde{\mathsf{TL}}(\beta)$ induces one on $\mathrm{Mod}_{\widetilde{\mathsf{TL}}}$ as follows:

$$\eta_{\overline{M},\overline{N}} : \mathsf{F}_{\overline{M}} \times_f \mathsf{F}_{\overline{N}} \to \mathsf{F}_{\overline{N}} \times_f \mathsf{F}_{\overline{M}} \tag{43}$$

$$(\eta_{\overline{M},\overline{N}})_k (a \otimes_{\overline{m},\overline{n}} (x \otimes y)) = a \circ \eta_{n,m} \otimes_{\overline{n},\overline{m}} (y \otimes x), \tag{44}$$

for all $x \in \overline{M}, y \in \overline{N}, a \in \mathrm{Hom}(n+m,k), k \in \widetilde{\mathsf{TL}}$, and extended linearly in the natural way. The components of $\eta$ are well-defined natural morphisms in $\mathrm{Mod}_{\widetilde{\mathsf{TL}}}$. Suppose indeed that $c \in \mathsf{TL}_{\overline{m}}, d \in \mathsf{TL}_{\overline{n}}, b \in \mathrm{Hom}(k,s)$. Then

$$(\eta_{\overline{M},\overline{N}})_s (ba \otimes_{\overline{m},\overline{n}} (cx \otimes dy)) = ba\eta_{n,m} \otimes_{\overline{n},\overline{m}} (dy \otimes cx)$$

$$= ba\eta_{n,m} (d \otimes_{\widetilde{\mathsf{TL}}} c) \otimes_{\overline{n},\overline{m}} (y \otimes x)$$

$$= ba(c \otimes_{\widetilde{\mathsf{TL}}} d)\eta_{n,m} \otimes_{\overline{n},\overline{m}} (y \otimes x)$$

$$= b(\eta_{\overline{M},\overline{N}})_k (a(c \otimes_{\widetilde{\mathsf{TL}}} d) \otimes_{\overline{m},\overline{n}} (x \otimes y)).$$

It is straightforward (but tedious) to show that $\eta_,$ is natural in both entries. (An example of such verification is done below for the twist $\theta$.) The check of the hexagonal axiom is the last step. Again any element of $((\mathsf{F}_{\overline{L}} \times_f \mathsf{F}_{\overline{M}}) \times_f \mathsf{F}_{\overline{N}})(k)$ is a linear combination of terms of the form $b \otimes_{\overline{l},\overline{m},\overline{n}} ((x \otimes y) \otimes z)$ where $b \in \mathrm{Hom}(l+m+n,k)$ for some $k$, and $x \in \overline{L}, y \in \overline{M}, z \in \overline{N}$. Then the upper part of the hexagon gives (we dropped the $k$ index to lighten the notation)

$$\alpha_{\overline{M},\overline{N},\overline{L}} \circ \eta_{\overline{L},\overline{M} \times_f \overline{N}} \circ \alpha_{\overline{L},\overline{M},\overline{N}} (b \otimes_{\overline{l},\overline{m},\overline{n}} ((x \otimes y) \otimes z))$$

$$= \alpha_{\overline{M},\overline{N},\overline{L}} \circ \eta_{\overline{L},\overline{M} \times_f \overline{N}} (b \otimes_{\overline{l},\overline{m},\overline{n}} (x \otimes (y \otimes z)))$$

$$= \alpha_{\overline{M},\overline{N},\overline{L}} (b\eta_{m+n,l} \otimes_{\overline{m},\overline{n},\overline{l}} ((y \otimes z) \otimes x))$$

$$= b\eta_{m+n,l} \otimes_{\overline{m},\overline{n},\overline{l}} (y \otimes (z \otimes x))$$

and the lower one

$$(1_{\overline{M}} \otimes \eta_{\overline{L},\overline{N}}) \circ \alpha_{\overline{M},\overline{L},\overline{N}} \circ (\eta_{\overline{L},\overline{M}} \otimes 1_{\overline{N}})(b \otimes_{\overline{l},\overline{m},\overline{n}} ((x \otimes y) \otimes z))$$
$$= (1_{\overline{M}} \otimes \eta_{\overline{L},\overline{N}}) \circ \alpha_{\overline{M},\overline{L},\overline{N}}(b(\eta_{m,l} \otimes 1_n) \otimes_{\overline{m},\overline{l},\overline{n}} ((y \otimes x) \otimes z)$$
$$= (1_{\overline{M}} \otimes \eta_{\overline{L},\overline{N}})(b(\eta_{m,l} \otimes 1_n) \otimes_{\overline{m},\overline{l},\overline{n}} (y \otimes (x \otimes z)))$$
$$= b(\eta_{m,l} \otimes 1_n)(1_m \otimes \eta_{n,l}) \otimes_{\overline{m},\overline{n},\overline{l}} (y \otimes (z \otimes x))$$
$$= b\eta_{m+n,l} \otimes_{\overline{m},\overline{n},\overline{l}} (y \otimes (z \otimes x)), \quad \text{by (1)}.$$

Since the two expressions coincide, the family of commutors $\eta_{\overline{M},\overline{N}}$, for $(\overline{m}, \overline{M}), (\overline{n}, \overline{N}) \in \text{Ob Mod}_{\widetilde{\text{TL}}}$, satisfies the first hexagon axiom. The proof of the second is similar and the above discussion establishes the braided structure of $\text{Mod}_{\widetilde{\text{TL}}}$.

**Proposition 3.3.** *The category* $\text{Mod}_{\widetilde{\text{TL}}}$ *together with the bifunctor* $\times_f$*, the identity* $I$*, the associator* $\alpha$*, the unitors* $\lambda, \rho$ *and the braiding* $\eta$ *is a braided category.*

## 3.2 Rigidity and $\text{Mod}_{\widetilde{\text{TL}}}$

The representation theory of rational conformal field theories offers examples of modular categories. These categories satisfy even more structural constraints than the braided tensor ones. This section identifies conditions under which the braided tensor categories $\text{Mod}_{\widetilde{\text{TL}}}$ can be endowed with some of these additional structures.

A braided category is rigid if, for every object $F$ in the category, there correpond two others, its right and left duals $F^*$ and $^*F$, and morphisms

$$e_F : F^* \otimes F \longrightarrow I \qquad \iota_F : I \longrightarrow F \otimes F^*, \tag{45}$$

$$e'_F : F \otimes^* F \longrightarrow I \qquad \iota'_F : I \longrightarrow^* F \otimes F, \tag{46}$$

such that the compositions

$$F \xrightarrow{\iota_F \otimes \text{id}_F} F \otimes F^* \otimes F \xrightarrow{\text{id}_F \otimes e_F} F \tag{47}$$

$$!F^* \xrightarrow{\text{id}_{F^*} \otimes \iota_F} F^* \otimes F \otimes F^* \xrightarrow{e_F \otimes \text{id}_{F^*}} F^* \tag{48}$$

are the identity morphism, on $F$ and $F^*$, respectively. If such morphisms exists, they are unique up to isomorphism. Similar axioms hold for $e'_F$ and $\iota'_F$. The rigidity axiom insures, in CFT, that any primary field $\phi$ has a right partner $\phi^*$ and a left one $^*\phi$ such that the correlation functions $\langle(\phi^*)\phi\rangle$ and $\langle\phi(^*\phi)\rangle$ are non-zero. Often the left and right partners coincide. Actually in all rigid braided category, the two duals are always isomorphic, since once can show that $(\iota_F)' \equiv \eta_{F,F^*}^{-1} \circ \iota_F : 1 \to F^* \otimes F$, and $(e_F)' \equiv e_F \circ \eta_{F,F^*} : 1 \to F \otimes F^*$, also satisfy the axioms for the right dual.

Note that the category $\widetilde{\text{TL}}$ is rigid, with $n^* \equiv {}^*n \equiv n$ for all objects $n \in \widetilde{\text{TL}}$, and duality morphisms

$$\bar{\iota}_m = \left. m \left\{ \vphantom{\Big|} \begin{matrix} \vdots \\ \vdots \end{matrix} \right. \right) \in \text{Hom}_{\widetilde{\text{TL}}}(0, 2m), \qquad \bar{e}_m = \left( \left. \begin{matrix} \vdots \\ \vdots \end{matrix} \vphantom{\Big|} \right\} m \right) \in \text{Hom}_{\widetilde{\text{TL}}}(2m, 0). \tag{49}$$

It is then straightforward to verify that $(1_m \otimes_{\widetilde{\text{TL}}} \bar{e}_m) \circ (\bar{\iota}_m \otimes_{\widetilde{\text{TL}}} 1_m) = (\bar{e}_m \otimes_{\widetilde{\text{TL}}} 1_m) \circ (1_m \otimes_{\widetilde{\text{TL}}} \bar{\iota}_m) = 1_m$.

The rigidity axiom is fulfilled for $\text{Mod}_{\widetilde{\text{TL}}}$ when the algebras $\text{TL}_n(\beta)$ are semisimple for all $n$, that is when $q$ is not a root of unity (recall that $\beta = -q - q^{-1}$). For these values of $\beta$, the

cellular modules $S_{n,k}$, $0 \le k \le n$ with $k = n \bmod 2$, form a complete set of non-isomorphic irreducible modules of $\mathsf{TL}_n$. All other (finite) modules of $\mathsf{TL}_n$ are direct sums of these. Their fusion product is known [3, 4].

**Proposition 3.4.** *If q is not a root of unity, then*

$$\mathsf{F}_m \times_f \mathsf{F}_n \simeq \bigoplus_{j=|m-n|}^{m+n}{}' \mathsf{F}_j, \qquad \text{with } \mathsf{F}_m \equiv \mathsf{F}_{m,S_{m,m}}$$

*and where $\oplus'$ stands for a direct sum with a step equal to 2.*

This simple fusion rule leads to the following result.

**Corollary 3.5.** *If q is not a root of unity, then*

$$\mathsf{F}_m \times_f \mathsf{F}_m \simeq \bigoplus_{j=0}^{2m}{}' \mathsf{F}_j \qquad \text{and} \qquad \text{Nat}(\mathsf{F}_0, \mathsf{F}_m \times_f \mathsf{F}_n) \simeq \text{Nat}(\mathsf{F}_m \times_f \mathsf{F}_n, \mathsf{F}_0) \simeq \delta_{m,n}\mathbb{C}$$

*as vector spaces.*

In other words $\mathsf{F}_m \times_f \mathsf{F}_n$ has a direct summand isomorphic to $\mathsf{I} = \mathsf{F}_0$ if and only if $m = n$. The left and right duals of $\mathsf{F}_m$ can thus be chosen to coincide with $\mathsf{F}_m$. With this observation, the construction of the four functorial morphisms $e_\mathsf{F}, \iota_\mathsf{F}, e'_\mathsf{F}$ and $\iota'_\mathsf{F}$ is straightforward. We detail that of $e_\mathsf{F}$ and $\iota_\mathsf{F}$. To define $\iota_m$, note first that

$$\bar{e}_m = \left( \vcenter{\hbox{}} \right.$$

is the only $(0, 2m)$-diagram whose tensor product with

$$\vcenter{\hbox{}} \otimes \vcenter{\hbox{}}$$

is non-zero. Thus define the components $\iota_m(k) : \mathsf{I}(k) \longrightarrow (\mathsf{F}_m \times_f \mathsf{F}_m^*)(k)$ of the morphism $\iota_{\mathsf{F}_m} \equiv \iota_m$ to be

$$f \otimes_\mathbb{C} 1 \longmapsto \alpha f \left( \vcenter{\hbox{}} \otimes_{m,m} \left( \vcenter{\hbox{}} \otimes \vcenter{\hbox{}} \right) \right.$$

where $f \in \text{Hom}(0, k)$ and $\alpha \in \mathbb{C}$ a constant to be fixed. A quick check shows that this is a morphism. (This morphism $\iota_m$ exists even for $q$ a root of unity.) Note that the definition of $\iota_m$ uses $\bar{e}_m \in \text{Hom}(2m, 0)$ and, as will be seen below, that of $e_m$ uses $\bar{\iota}_m \in \text{Hom}(0, 2m)$, a fact that could be confusing.

The definition of $e_m : \mathsf{F}_m \times_f \mathsf{F}_m \to \mathsf{I}$ requires the primitive idempotent $wj_m \in \mathsf{TL}_m$, known as the Wenzl-Jones projector. It is the unique non-zero element of $\mathsf{TL}_m$ such that $wj_m \cdot wj_m = wj_m$ and $wj_m \cdot e_i = e_i \cdot wj_m = 0$, $1 \le i < m$ [16]. Thus $\mathbb{C} \cdot wj_m \simeq S_{m,m}$ as a left module. Such idempotents $wj_m$ exist for all $m$ if and only if $q$ is not a root of unity. Moreover

$$wj_m \cdot \vcenter{\hbox{}} = \vcenter{\hbox{}}$$

in $S_{m,m}$. The components $e_m(k) : (\mathsf{F}_m \times_f \mathsf{F}_m)(k) \to \mathsf{I}(k)$ of $e_m$ are linear maps defined by

$$g \otimes_{m,m} \left( wj_m \cdot \vcenter{\hbox{}} \otimes wj_m \cdot \vcenter{\hbox{}} \right) \longmapsto \alpha' g \circ (wj_m \otimes_{\widetilde{\mathsf{TL}}} wj_m) \vcenter{\hbox{}} \otimes 1$$

for $g \in \mathrm{Hom}(2m, k)$ some $\alpha' \in \mathbb{C}$. The two constants $\alpha$ and $\alpha'$ are tied by the rigidity axioms (47) and (48). For example (47) gives for $f \in \mathrm{Hom}(m, k)$

$$(\mathrm{id}_m \otimes_{\widetilde{\mathrm{TL}}} e_m)(\iota_m \otimes_{\widetilde{\mathrm{TL}}} \mathrm{id}_m)(f \otimes_{0,m} (1 \otimes \boxed{\cdots}\,))$$

$$= \alpha(\mathrm{id}_m \otimes_{\widetilde{\mathrm{TL}}} e_m)\Big(f \circ \;\Big) \otimes_{m,m,m} \big(\boxed{\cdots} \otimes \boxed{\cdots} \otimes \boxed{\cdots}\big)$$

$$= \alpha\alpha'\Big(f \circ \;\boxed{wj_m \cdot}\;\Big) \otimes_m \boxed{\cdots}\;\Big)$$

$$= \alpha\alpha'\big(f \circ wj_m \otimes_m \boxed{\cdots}\big) = \alpha\alpha'\big(f \otimes_m \boxed{\cdots}\big)$$

and $(\mathrm{id}_m \otimes_{\widetilde{\mathrm{TL}}} e_m) \circ (\iota_m \otimes_{\widetilde{\mathrm{TL}}} \mathrm{id}_m)$ is the identity if and only if $\alpha\alpha' = 1$. The second axiom (48) does not add any constraints and, if $\alpha$ and $\alpha'$ are chosen to be 1, $e$ and $\iota$ satisfy all the axioms. The constructions and verifications of this section and the previous one prove the following result.

**Proposition 3.6.** *If $q$ is not root of unity, the category* $\mathrm{Mod}_{\widetilde{\mathrm{TL}}}$ *together with the bifunctor* $\times_f$, *the identity* $\mathsf{I}$, *the associator* $\alpha$, *the unitors* $\lambda, \rho$, *the braiding* $\eta$ *and the morphisms* $e, \iota, e'$ *and* $\iota'$, *is rigid.*

A *ribbon category* is a rigid category endowed with a functorial isomorphism $\theta$ and a dual isomorphism $\theta^*$ satisfying $(\theta_\mathsf{F})^* = \theta_{\mathsf{F}^*}$ and $^*(\theta_\mathsf{F}) = \theta_{^*\mathsf{F}}$. The isomorphism $\theta$ is called a *twist*. The duals are defined by the composition (omitting associators and unitors)

$$(\theta_\mathsf{F})^* : \mathsf{F}^* \xrightarrow{\;\mathrm{id}_{\mathsf{F}^*} \otimes \iota_\mathsf{F}\;} \mathsf{F}^* \otimes \mathsf{F} \otimes \mathsf{F}^* \xrightarrow{\;\mathrm{id}_{\mathsf{F}^*} \otimes \theta_\mathsf{F} \otimes \mathrm{id}_{\mathsf{F}^*}\;} \mathsf{F}^* \otimes \mathsf{F} \otimes \mathsf{F}^* \xrightarrow{\;e_\mathsf{F} \otimes \mathrm{id}_{\mathsf{F}^*}\;} \mathsf{F}^*,$$

and similarly for the left duals. The natural isomorphism $\theta$ for $\mathrm{Mod}_{\widetilde{\mathrm{TL}}}$ will first be constructed and then the compatibility between duals checked.

The twist on $\widetilde{\mathrm{TL}}$ induces a twist on $\mathrm{Mod}_{\widetilde{\mathrm{TL}}}$ as follows. For $c \in \mathrm{Hom}(m, k)$ and $x \in \overline{\mathsf{M}}$, define $\theta_{\mathsf{F}_{\overline{\mathsf{M}}}}(k) = \theta_{\overline{\mathsf{M}}}(k)$ by

$$c \otimes_{\overline{m}} x \longmapsto \theta_{\overline{\mathsf{M}}}(k)(c \otimes_{\overline{m}} x) = c\theta_m \otimes_{\overline{m}} x. \tag{50}$$

Since the elements $\theta_m \in \mathrm{TL}_m$ are central, $\theta_{\overline{\mathsf{M}}}(k)$ is well-defined. Since its (unique) eigenvalue on an indecomposable module is never zero, $\theta_{\overline{\mathsf{M}}}(k)$ is also invertible. The next step is to prove that it is a natural transformation of the identity functor of $\mathrm{Mod}_{\widetilde{\mathrm{TL}}}$; to see this, one must prove that for all functors $\mathsf{F}_{\overline{\mathsf{N}}}, \mathsf{F}_{\overline{\mathsf{M}}} \in \mathrm{Mod}_{\widetilde{\mathrm{TL}}}$, and all natural transformation $\mu : \mathsf{F}_{\overline{\mathsf{N}}} \to \mathsf{F}_{\overline{\mathsf{M}}} \in \mathrm{Hom}_{\mathrm{Mod}_{\widetilde{\mathrm{TL}}}}$, the components of $(\mu \circ \theta_{\overline{\mathsf{N}}})$ and $(\theta_{\overline{\mathsf{M}}} \circ \mu)$ must be equal on all objects $k \in \widetilde{\mathrm{TL}}$. Let $a \otimes_{\widetilde{\mathrm{TL}}} x$ be some element of $\mathsf{F}_{\overline{\mathsf{N}}}(k)$, so $a \in \mathrm{Hom}(n, k)$, $x \in \overline{\mathsf{N}}$, and write $\mu_k(a \otimes_{\widetilde{\mathrm{TL}}} x) \equiv \sum_i b_i \otimes_{\widetilde{\mathrm{TL}}} y_i$, where the sum is over some finite set, the $b_i$ are in $\mathrm{TL}_m$ and the $y_i$ in $\overline{\mathsf{M}}$. One then verifies that

$$\begin{aligned}
(\mu \circ \theta_{\overline{\mathsf{N}}})_k(a \otimes_{\widetilde{\mathrm{TL}}} x) &= \mu_k(a \circ \theta_n \otimes_{\widetilde{\mathrm{TL}}} x) \\
&= \mu_k(\theta_k \circ a \otimes_{\widetilde{\mathrm{TL}}} x), \quad \text{by (34)} \\
&= \theta_k \mu_k(a \otimes_{\widetilde{\mathrm{TL}}} x) = \theta_k \sum_i b_i \otimes y_i \\
&= \sum_i b_i \theta_m \otimes_{\widetilde{\mathrm{TL}}} y_i, \quad \text{again by (34)} \\
&= (\theta_{\overline{\mathsf{M}}} \circ \mu)_k(a \otimes_{\widetilde{\mathrm{TL}}} x),
\end{aligned}$$

where we used the fact that $\theta$ is a natural transformation on $\widetilde{\mathsf{TL}}$ and $\mu$ is natural on $\mathrm{Mod}_{\widetilde{\mathsf{TL}}}$. Since $(\mu \circ \theta_{\overline{\mathsf{N}}})_k$ and $(\theta_{\overline{\mathsf{M}}} \circ \mu)$ are both linear, it follows that they must be equal on $\mathsf{F}_{\overline{\mathsf{N}}}(k)$. Finally, let $a \in \mathrm{Hom}(m+n, k)$ and $x \in \overline{\mathsf{M}}$ and $y \in \overline{\mathsf{N}}$ such that $a \otimes_{\overline{m},\overline{n}} (x \otimes y) \in \mathsf{F}_{\overline{\mathsf{M}} \times_f \overline{\mathsf{N}}}$. Then

$$
\theta_{\overline{\mathsf{M}} \times_f \overline{\mathsf{N}}}(a \otimes_{\overline{m},\overline{n}} (x \otimes y)) = a\theta_{m+n} \otimes_{\overline{m},\overline{n}} (x \otimes y)
$$

$$
= (a \circ \eta_{n,m} \circ \eta_{m,n} \circ (\theta_m \otimes_{\widetilde{\mathsf{TL}}} \theta_n)) \otimes_{\overline{m},\overline{n}} (x \otimes y), \quad \text{since } \theta \text{ is a twist on } \widetilde{\mathsf{TL}}
$$

$$
= (a \circ (\theta_m \otimes_{\widetilde{\mathsf{TL}}} \theta_n) \circ \eta_{n,m} \circ \eta_{m,n}) \otimes_{\overline{m},\overline{n}} (x \otimes y), \quad \text{by 2.4}
$$

$$
= \eta_{\overline{\mathsf{N}},\overline{\mathsf{M}}}(a \circ (\theta_m \otimes_{\widetilde{\mathsf{TL}}} \theta_n) \circ \eta_{n,m} \otimes_{\overline{n},\overline{m}} (y \otimes x))
$$

$$
= \eta_{\overline{\mathsf{N}},\overline{\mathsf{M}}} \circ \eta_{\overline{\mathsf{M}},\overline{\mathsf{N}}}(a \circ (\theta_m \otimes_{\widetilde{\mathsf{TL}}} \theta_n) \otimes_{\overline{m},\overline{n}} (x \otimes y))
$$

$$
= \eta_{\overline{\mathsf{N}},\overline{\mathsf{M}}} \circ \eta_{\overline{\mathsf{M}},\overline{\mathsf{N}}} \circ (\theta_{\overline{\mathsf{M}}} \otimes_{\widetilde{\mathsf{TL}}} \theta_{\overline{\mathsf{N}}})(a \otimes_{\overline{m},\overline{n}} (x \otimes y)),
$$

and the twist $\theta$ on $\mathrm{Mod}_{\widetilde{\mathsf{TL}}}$ verifies the axiom (26).

It simply remains to verify that the twist is compatible with the duals; we show that it is compatible with right duals, as the proof for the left ones is very similar. First note that for all $1 \le i \le m$,

$$
t_i \bar{\iota}_m = t_{2m-i} \bar{\iota}_m, \qquad \bar{e}_m t_i = \bar{e}_m t_{2m-i},
$$

where $\bar{\iota}_m, \bar{e}_m$ are the duality morphisms from $\widetilde{\mathsf{TL}}$, introduced in equation (49). It therefore follows that

$$
(\theta_m \otimes_{\widetilde{\mathsf{TL}}} 1_m)\iota_m = (1_m \otimes_{\widetilde{\mathsf{TL}}} \theta_m)\iota_m \qquad \text{and} \qquad e_m(\theta_m \otimes_{\widetilde{\mathsf{TL}}} 1_m) = e_m(1_m \otimes_{\widetilde{\mathsf{TL}}} \theta_m)
$$

where $e_m$ and $\iota_m$ are now the components of $e_{\mathsf{F}}$ and $\iota_{\mathsf{F}}$. Using this observation with the definition of the twist and duals in $\mathrm{Mod}_{\widetilde{\mathsf{TL}}}$, one quickly sees that for all $\mathsf{F} \in \mathrm{Mod}_{\widetilde{\mathsf{TL}}}$

$$
(\theta_{\mathsf{F}} \otimes 1)\iota_{\mathsf{F}} = (1 \otimes \theta_{\mathsf{F}})\iota_{\mathsf{F}} \qquad \text{and} \qquad e_{\mathsf{F}}(\theta_{\mathsf{F}} \otimes 1) = e_{\mathsf{F}}(1 \otimes \theta_{\mathsf{F}}).
$$

The right dual of $\theta_{\mathsf{F}}$ is thus

$$
(\theta_{\mathsf{F}})^* = (e_{\mathsf{F}} \otimes \mathrm{id}_{\mathsf{F}}) \circ [(\mathrm{id}_{\mathsf{F}} \otimes \theta_{\mathsf{F}} \otimes \mathrm{id}_{\mathsf{F}}) \circ (\mathrm{id}_{\mathsf{F}} \otimes \iota_{\mathsf{F}})]
$$

$$
= [(e_{\mathsf{F}} \otimes \mathrm{id}_{\mathsf{F}}) \circ (\mathrm{id}_{\mathsf{F}} \otimes \mathrm{id}_{\mathsf{F}} \otimes \theta_{\mathsf{F}})] \circ (\mathrm{id}_{\mathsf{F}} \otimes \iota_{\mathsf{F}})
$$

$$
= (\mathrm{id}_{\mathsf{I}} \otimes \theta_{\mathsf{F}}) \circ [(e_{\mathsf{F}} \otimes \mathrm{id}_{\mathsf{F}}) \circ (\mathrm{id}_{\mathsf{F}} \otimes \iota_{\mathsf{F}})]
$$

$$
= (\mathrm{id}_{\mathsf{I}} \otimes \theta_{\mathsf{F}}) = \theta_{\mathsf{F}}
$$

where we have used the fact that $\mathsf{F}^* \equiv \mathsf{F}$ and, to get the last line, that $\mathrm{Mod}_{\widetilde{\mathsf{TL}}}$ is rigid. The twist $\theta$ in $\mathrm{Mod}_{\widetilde{\mathsf{TL}}}$ is thus compatible with its duals.

These checks on $\theta$ holds for any $q$. However, for a category to be a ribbon category, it needs to be rigid and thus, for $\mathrm{Mod}_{\widetilde{\mathsf{TL}}}$, $q$ may not be a root of unity.

**Proposition 3.7.** *If $q$ is not root of unity, the data $(\mathrm{Mod}_{\widetilde{\mathsf{TL}}}, \times_f, \mathsf{I}, \alpha, \lambda, \rho, \eta, e, \iota, e', \iota', \theta)$ define a ribbon category.*

Note that we also proved that the twist in $\widetilde{\mathsf{TL}}$ is also compatible with its duals, so $\widetilde{\mathsf{TL}}$ (with the appropriate data) is also a ribbon category.

The categories appearing naturally in minimal conformal field theories are the modular tensor categories. Beside being ribbon categories, the modular ones require among other things that all objects can be written as a finite direct sum of simple objects and that the number of (isomorphic classes of) simple objects be finite. (An object $A$ in an abelian category $\mathcal{C}$ is simple if any injective morphism $B \to A$ is either 0 or an isomorphism.) When $q$ is not a root of unity, the simple objects are the standard modules $\mathsf{S}_{n,k}$, $n \in \mathbb{N}_0$, and $0 \le k \le n$

with $n = k \bmod 2$. Corollary 3.2 shows that $\mathsf{F}_{n,\mathsf{S}_{n,n}}, n \in \mathbb{N}_0$, are all simple objects and non-isomorphic. However their number is infinite and the ribbon category $\mathrm{Mod}_{\overline{\mathsf{TL}}}$ cannot be a modular one.

The category $\mathrm{Mod}_{\overline{\mathsf{TL}}}$ cannot be rigid when $q$ is a root of unity. Indeed rigidity would imply the exactness of the bifunctor $\times_f$ which it is *not* when $q$ is a root of unity [4]. But it turns out that $\times_f$ is closed when restricted to projective modules, even when $q$ is a root of unity. It is thus natural to consider the full subcategory $\mathrm{Mod}_{\overline{\mathsf{TL}}}^{\mathrm{proj}}$ with objects restricted to projective modules (or, more precisely, functors of the form $\mathsf{F}_{n,\mathsf{P}}$ where P is projective $\mathsf{TL}_n$-module). The fusion coefficients are more intricate than those of Corollary 3.5. (See Proposition 4.1.1 of [3] or Section 4.1 and the quick computational tool 5.4 of [4].) However the Corollary's key feature, the one that allows for the introduction of morphisms $e_\mathsf{F}, e'_\mathsf{F}, \iota_\mathsf{F}$ and $\iota'_\mathsf{F}$, still holds:

$$\mathrm{Nat}(\mathsf{F}_0, \mathsf{F}_m \times_f \mathsf{F}_n) \simeq \mathrm{Nat}(\mathsf{F}_m \times_f \mathsf{F}_n, \mathsf{F}_0) \simeq \delta_{m,n}\mathbb{C}$$

as vector spaces, when $q$ is a root of unity such that the smallest positive integer $\ell$ such that $q^{2\ell} = 1$ is larger than 2. Here $\mathsf{F}_n$ stands now for $\mathsf{F}_{n,\mathsf{P}_{n,n}}$. The morphisms (45) and (46) thus exist. We have checked on a few cases that the rigidity conditions (47) hold for proper choices of the $e$'s and $\iota$'s. The full subcategory $\mathrm{Mod}_{\overline{\mathsf{TL}}}^{\mathrm{proj}}$ has an unexpected feature however: it is not any more an *abelian* category, a characteristic that is usually assumed in the study of fusion. Recall that, in an abelian category, every morphism has a kernel and a cokernel in the category. But, when a projective module $\mathsf{P}_{n,k}$ has three or four composition factors, there are morphisms $\mathsf{P}_{n,k} \to \mathsf{P}_{n,k}$ whose kernels and cokernels are not projective and thus absent from the subcategory $\mathrm{Mod}_{\overline{\mathsf{TL}}}^{\mathrm{proj}}$.

### 3.3   The monodromy $\eta_{\mathsf{N},\mathsf{M}} \circ \eta_{\mathsf{M},\mathsf{N}}$

Since the commutor $\eta$ is a natural isomorphism, the composition $\eta_{\mathsf{F}_{\overline{\mathsf{N}}},\mathsf{F}_{\overline{\mathsf{M}}}} \circ \eta_{\mathsf{F}_{\overline{\mathsf{M}}},\mathsf{F}_{\overline{\mathsf{N}}}}$ is a natural isomorphism of $\mathsf{F}_{\overline{\mathsf{M}} \times_f \overline{\mathsf{N}}}$ onto itself. In conformal field theories the eigenvalues of this isomorphism are related to the monodromy of correlation functions of primary fields. We shall refer to this map as the *monodromy* as in the study of modular tensor categories. Because of the axiom (26), the monodromy is completely determined by the twist (see Proposition 3.8 below). But the definition of the twist (50) shows that the definition of $\theta_{\mathsf{F}_{m,\overline{M}}}(k)$ depends only on $m$ and not on the component $k$. Moreover every $\mathsf{TL}_m$-module M is the component $m$ of a functor in $\mathrm{Ob}(\mathrm{Mod}_{\overline{\mathsf{TL}}})$. We thus restrict our study of the monodromy to the fusion of the $\mathsf{TL}_m$- and $\mathsf{TL}_n$-modules M and N.

The nature of the monodromy is easy to describe in the semisimple case, i.e. when $q$ is not a root of unity. However the morphism may have a nilpotent part when $q$ is a root of unity, as will be seen below. In the following we use fairly standard notations, writing $\mathsf{I}_{n,k}, \mathsf{S}_{n,k}$ and $\mathsf{P}_{n,k}$ for the irreducible, standard and projective modules over $\mathsf{TL}_n$. The standard $\mathsf{S}_{n,k}$ was described at the beginning of 3.1, the irreducible $\mathsf{I}_{n,k}$ is its irreducible quotient and $\mathsf{P}_{n,k}$ the projective cover of $\mathsf{I}_{n,k}$. (See [4,14], and also [1] where $\mathsf{S}_{n,k}$ is denoted by $W_k(n)$.)

The twist $\theta$ defines isomorphisms of modules over the Temperley-Lieb algebras. Indeed they are defined through the invertible central elements $\theta_n$ of $\mathsf{TL}_n$ and thus define isomorphisms of modules by left multiplication. The defining property (26) of the twist $\theta$ gives a rather explicit expression for the monodromy. We choose to state this (obvious) fact in a proposition to underline its crucial character and ease further references.

**Proposition 3.8.** *The monodromy $\eta_{\mathsf{N},\mathsf{M}} \circ \eta_{\mathsf{M},\mathsf{N}}$ is expressed in terms of the twist as*

$$\eta_{\mathsf{N},\mathsf{M}} \circ \eta_{\mathsf{M},\mathsf{N}} = \theta_{\mathsf{M} \times_f \mathsf{N}}(\theta_\mathsf{M} \otimes \theta_\mathsf{N})^{-1}. \tag{51}$$

When $q$ is generic (not a root of unity), then the algebras $\mathsf{TL}_n$ are semisimple for all $n$ and the standard modules $\mathsf{S}_{n,k}$, with $0 \le k \le n$ and $k \equiv n \bmod 2$, provide a complete list of non-isomorphic irreducible modules. Their fusion was given in terms of the associated functor in Proposition 3.4. Here is a simpler statement in terms of the modules themselves.

**Proposition 3.9** ( [3, 4]). *Let $n_1, n_2 \ge 1$ and $k_1, k_2$ such that $0 \le k_i \le n_i$ and $k_i \equiv n_i \bmod 2$. Then*

$$\mathsf{S}_{n_1,k_1} \times_f \mathsf{S}_{n_2,k_2} \simeq \bigoplus_{k=|k_1-k_2|}^{k_1+k_2}{}' \mathsf{S}_{n_1+n_2,k}$$

*when $q$ is not a root of unity. Again $\oplus'$ indicates a direct sum whose index step is two.*

3.8 then gives a complete characterization of the monodromy of standard modules in this generic case.

**Proposition 3.10.** *The monodromy $\eta_{\mathsf{S}_{n_2,k_2},\mathsf{S}_{n_1,k_1}} \circ \eta_{\mathsf{S}_{n_1,k_1},\mathsf{S}_{n_2,k_2}}$ acts on the direct summand $\mathsf{S}_{n_1+n_2,k}$ of $\mathsf{S}_{n_1,k_1} \times_f \mathsf{S}_{n_2,k_2}$ as the identity times*

$$\mu_{k_1,k_2,k} = q^{k(k/2+1)-k_1(k_1/2+1)-k_2(k_2/2+1)} \tag{52}$$

*when $q$ is not a root of unity.*

*Proof.* The central elements $c_n = \theta_n$ act as multiples of the identity on the irreducible $\mathsf{S}_{n,k}$ by Schur's lemma. The eigenvalues $\gamma_{n,k}$ of $c_n$ on the standard modules $\mathsf{S}_{n,k}$ are obtained in A.2. Thus

$$(\theta_{n_1} \otimes \theta_{n_2})^{-1}\big|_{\mathsf{S}_{n_1,k_1} \times_f \mathsf{S}_{n_2,k_2}} = \frac{1}{\gamma_{n_1,k_1}\gamma_{n_2,k_2}} \cdot \mathrm{id}\,.$$

The restriction of the monodromy to the direct summand $\mathsf{S}_{n_1+n_2,k}$ of the fusion is then $\gamma_{n_1+n_2,k}/(\gamma_{n_1,k_1}\gamma_{n_2,k_2})$. A direct computation leads to the desired expression. $\qquad\square$

It is worthwhile to note that the multiple of the identity is independent of $n_1$ and $n_2$ whose only role here is to fix the parities of $k_1$ and $k_2$.

When $q$ is a root of unity, the monodromy $\eta_{\mathsf{N,M}} \circ \eta_{\mathsf{M,N}}$ is still given by 3.8, but it might not be a multiple of the identity. Indeed the action of the central elements $c_n$ on $\mathsf{TL}_n$-modules is in general not such a multiple. The A.3 and the paragraph leading to it show that such a non-trivial action, i.e. a non-diagonalisable action, might occur only on projective modules with three or four composition factors. Let $\mathsf{P}_{n.k}$ be such a projective module. Then $\mathrm{Hom}(\mathsf{P}_{n,k},\mathsf{P}_{n,k})$ is two-dimensional. Beside the identity id, there is a map sending the head of $\mathsf{P}_{n,k}$ to its socle, both being isomorphic to $\mathsf{I}_{n,k}$. Let $f$ be this map, that is, a map such that $\mathrm{im}\, f$ is the socle of $\mathsf{P}_{n,k}$. This map is nilpotent: $f^2 = 0$. We now give two examples of such non-trivial action of the monodromy.

One of the simplest cases occurs when $q^{2\ell} = 1$ for $\ell = 3$. Then, even though the standard modules $\mathsf{S}_{2,2}$ and $\mathsf{S}_{1,1}$ are irreducible, their fusion $\mathsf{S}_{2,2} \times_f \mathsf{S}_{1,1}$ is not. Using the expressions computed in [3, 4], one finds $\mathsf{S}_{2,2} \times_f \mathsf{S}_{1,1} \simeq \mathsf{P}_{3,3}$ where $\mathsf{P}_{3,3}$ is an indecomposable projective module. This projective is three-dimensional, has three one-dimensional composition factors: $\mathsf{I}_{3,1}$ once and $\mathsf{I}_{3,3}$ twice. The latter composition factors are isomorphic to the socle and head of the module. Hence, even though $\theta_2$ and $\theta_1$ act as multiples of the identity on $\mathsf{S}_{2,2}$ and $\mathsf{S}_{1,1}$, the morphism defined by $c_3$ is not diagonalisable on $\mathsf{P}_{3,3}$. In fact a direct computation shows that the monodromy in this case is $\eta_{\mathsf{S}_{1,1},\mathsf{S}_{2,2}} \circ \eta_{\mathsf{S}_{2,2},\mathsf{S}_{1,1}} = e^{4\pi i/3} \cdot \mathrm{id} + \nu f$ if $q$ is chosen to be $e^{2\pi i/3}$. Here $\nu$ is a non-zero constant (that depends on the basis) and $f$ is the map described above. Note that the (unique) eigenvalue of the monodromy is still correctly predicted by (52), as it should be: $\mu_{2,1,3} = q^2 = e^{4\pi i/3}$.

Our second example is more intricate: we shall study the monodromy on the product $\mathsf{TL}_2 \times_f \mathsf{TL}_2$ for $q$ generic and $q = \sqrt{-1}$. If $q$ is generic, then $\mathsf{TL}_2 \simeq \mathsf{S}_{2,0} \oplus \mathsf{S}_{2,2}$. The linearity of the fusion together with 3.9 gives

$$\mathsf{TL}_2 \times_f \mathsf{TL}_2 \simeq \mathsf{TL}_4 \simeq \mathsf{S}_{4,0} \oplus \mathsf{S}_{4,0} \oplus \mathsf{S}_{4,2} \oplus \mathsf{S}_{4,2} \oplus \mathsf{S}_{4,2} \oplus \mathsf{S}_{4,4}.$$

Since $q$ is generic, the monodromy $\eta_{\mathsf{TL}_2,\mathsf{TL}_2} \circ \eta_{\mathsf{TL}_2,\mathsf{TL}_2}$ is diagonalisable with eigenvalues given by 3.10. The eigenvalues on the two copies isomorphic to $\mathsf{S}_{4,0}$ are $q^{-8}$ and 1, on the three $\mathsf{S}_{4,2}$ they are $q^{-4}$, 1 and 1, and on $\mathsf{S}_{4,4}$ it is $q^4$. (Note that $\eta_{\mathsf{TL}_2,\mathsf{TL}_2} \circ \eta_{\mathsf{TL}_2,\mathsf{TL}_2}$ does not take the same eigenvalues on isomorphic copies of the standard modules in $\mathsf{TL}_4$, since the multiple $\mu_{k_1,k_2,k}$ depends also on the modules begin fused.) If $q = \sqrt{-1}$ and thus $\beta = 0$, then $\mathsf{TL}_2 \simeq \mathsf{P}_{2,2}$ and the fusion is then

$$\mathsf{TL}_2 \times_f \mathsf{TL}_2 \simeq \mathsf{P}_{2,2} \times_f \mathsf{P}_{2,2} \simeq \mathsf{P}_{4,2} \oplus \mathsf{P}_{4,2} \oplus \mathsf{P}_{4,4}.$$

At this value of $q$, the isomorphism $\eta_{\mathsf{TL}_2,\mathsf{TL}_2} \circ \eta_{\mathsf{TL}_2,\mathsf{TL}_2}$ is even more complicated because none of the three isomorphisms in $\theta_{\mathsf{TL}_4}(\theta_{\mathsf{TL}_2} \otimes \theta_{\mathsf{TL}_2})^{-1}$ is a multiple of the identity on the modules they act upon. Still it has a unique eigenvalue as $\mu_{2,2,0} = \mu_{2,2,2} = \mu_{2,2,4} = 1$. While $\eta_{\mathsf{TL}_2,\mathsf{TL}_2} \circ \eta_{\mathsf{TL}_2,\mathsf{TL}_2}$ is non-diagonalisable, it is possible to find two subspaces A and B, both isomorphic to $\mathsf{P}_{4,2}$ that allows for an easy description of the morphism. Let C be the other summand $\mathsf{P}_{4,4}$. Then $\delta = \eta_{\mathsf{TL}_2,\mathsf{TL}_2} \circ \eta_{\mathsf{TL}_2,\mathsf{TL}_2}$ can be broken down into its action on each summand as

$$\begin{bmatrix} \delta_{\mathsf{A,A}} \\ \delta_{\mathsf{A,B}} \\ \delta_{\mathsf{A,C}} \end{bmatrix} : \mathsf{A} \longrightarrow \mathsf{TL}_4, \qquad \begin{bmatrix} \delta_{\mathsf{B,A}} \\ \delta_{\mathsf{B,B}} \\ \delta_{\mathsf{B,C}} \end{bmatrix} : \mathsf{B} \longrightarrow \mathsf{TL}_4, \qquad \begin{bmatrix} \delta_{\mathsf{C,A}} \\ \delta_{\mathsf{C,B}} \\ \delta_{\mathsf{C,C}} \end{bmatrix} : \mathsf{C} \longrightarrow \mathsf{TL}_4.$$

They are

$$\begin{aligned} \delta_{\mathsf{A,A}} &= \mathrm{id} + \sigma f, & \delta_{\mathsf{A,B}} &= \mathrm{id}_{\mathsf{A,B}}, & \delta_{\mathsf{A,C}} &= 0, \\ \delta_{\mathsf{B,A}} &= 0, & \delta_{\mathsf{B,B}} &= \mathrm{id} + \nu f, & \delta_{\mathsf{B,C}} &= 0, \\ \delta_{\mathsf{C,A}} &= 0, & \delta_{\mathsf{C,B}} &= 0, & \delta_{\mathsf{C,C}} &= \mathrm{id} + \rho f, \end{aligned}$$

where $\sigma, \nu, \rho$ are non-zero constants and $\mathrm{id}_{\mathsf{A,B}}$ stands for the isomorphism between A and B. From these maps, it is straighforward to compute the Jordan form of $\eta$. Its non-trivial Jordan blocks are 2 blocks $3 \times 3$ and 2 blocks $2 \times 2$.

The root $q = \sqrt{-1}$ is somewhat special in the representation theory of the algebra $\mathsf{TL}_n$: It is the only value for which the semisimplicity of $\mathsf{TL}_n$ varies with the parity of $n$. (For all other roots $q^{2\ell} = 1$ with $\ell \geq 3$, the algebra $\mathsf{TL}_n(\beta = -q - q^{-1})$, $n \geq \ell$, is never semisimple.) Although the example above was given at this particular value $q = \sqrt{-1}$, it seems to be representative of what happens at other values of $q$.

## 4  Braiding and integrability

One of the most profound uses of Temperley-Lieb algebras in physics is in the study of solvable models, like the XXZ Hamiltonians or loop models on two-dimensional lattices. The goal of the present section is to tie braiding and integrability in some of these statistical models. The former will appear through the *elementary brainding* $\eta_{1,1}$ (or $t_i(n)$) that was used in 2.6 to write all other components $\eta_{r,s}$ of the braiding natural isomorphism. The latter will also be cast in terms of a fundamental "face operator" that must satisfy three identities. The physical object, that is, the Hamiltonian or the transfer matrix, is then defined in terms of several copies of this face operator.

In the literature on statistical models, the *face operator* $X_i(q, u)$ is also an element of one of the algebras $\mathsf{TL}_n(\beta)$. It depends on several parameters: The *spectral parameter* $\lambda$, tied to $\beta$ by $\beta = -q - q^{-1}$ and $q = e^{i\lambda}$, and the *anisotropy parameter* $u$ that measures the ratio of the interaction constants along two linearly independent vectors spanning the lattice. As for the $t_i$, the $X_i$ is usually a linear combination of $\mathsf{TL}_n$ generators and it is represented graphically by

$$\overline{\diamondsuit u \diamondsuit}_{\substack{i \\ i+1}}$$

Since all faces will be evaluated at the same value of the parameter $q$ (or $\lambda$), this parameter is often omitted. In terms of the face $X_i(q, u)$, the transfer matrix $D_n(\lambda, u) \in \mathrm{Hom}(n, n)$ on $n$ sites is constructed out of $2n$ tiles organized in diagonal lines. For example the case $n = 3$ is depicted as follows:

$$D_3(\lambda, u) \;=\; \text{(diagram)} \;.$$

In the notation of the previous section, it is thus

$$D_n(\lambda, u) = (1_n \otimes z^t) \circ \left( \prod_{i=1}^{n} X_i(q, u) \prod_{i=n}^{1} X_i(q, u) \right) \circ (1_n \otimes z) \in \mathrm{Hom}(n, n). \tag{53}$$

Its physical properties are revealed through its spectrum in some representations. It was recognized by Behrend, Pearce and O'Brien [17] that some algebraic conditions on the face operator $X_i(q, u)$ ensure that the transfer matrix, constructed from it, will have the properties that $D_n(\lambda, u) \circ D_n(\lambda, v) - D_n(\lambda, v) \circ D_n(\lambda, u) = 0$ in $\mathsf{TL}_n$. This means that, in any representation $\phi : \mathsf{TL}_n \to gl(\mathsf{V})$ with $\mathsf{V}$ some vector space, the matrices $\phi(D_n(\lambda, u))$ and $\phi(D_n(\lambda, v))$ will commute for all values $u$ and $v$. The modes $\phi(D_n(\lambda, u))$ in any expansion with respect to $u$ (Taylor's expansion, Fourier's, ...) will commute, that is, they will be integrals of motions. Thus the integrability of the models based on such a transfer matrix $D_n$ follows from these algebraic conditions. Here they are.

**Proposition 4.1** (section 3.4 of [17]). *If $X_i(q, u)$ verifies the following three conditions, then $D_n(\lambda, u) \circ D_n(\lambda, v) = D_n(\lambda, v) \circ D_n(\lambda, u)$, for all $u, v \in \mathbb{C}$:*

$$\text{(Yang-Baxter equation)} \quad X_i(q, u) X_{i+1}(q, v) X_i(q, v/u) = X_{i+1}(q, v/u) X_i(q, v) X_{i+1}(q, u), \tag{54}$$

$$\text{(inversion relation)} \quad X_i(q, u) X_i(q, u^{-1}) = \rho(q, u)\, \mathrm{id}, \tag{55}$$

$$\text{(boundary Yang-Baxter)} \quad X_i(q, u) X_{i+1}(q, v) \circ (z \otimes z)$$
$$= X_i(q, u) X_{i-1}(q, v) \circ (z \otimes z) \tag{56}$$

*for some non-identically zero function $\rho(q, u)$.*

These conditions are found in the literature drawn as follows:

(Yang-Baxter equation)

(inversion relation) $\quad = \rho(q, u)\, \mathrm{id}$

(boundary Yang-Baxter)

It is not too difficult to construct such a face operator $X_i$ out of the elementary braiding element $\eta_{1,1} = t_i$. As an intermediary step, consider

$$y_i(u) = u^{-1}t_i - ut_i^{-1}.$$

Both products $y_i(u)y_{i+1}(v)y_i(w)$ and $y_{i+1}(w)y_i(v)y_{i+1}(u)$ contain eight terms, each cubic in the generators $t_i$, $t_{i+1}$ and their inverses. The identity (16) gives rise to the six following ones:

$$t_i t_{i+1} t_i = t_{i+1} t_i t_{i+1}, \qquad t_{i+1} t_i t_{i+1}^{-1} = t_i^{-1} t_{i+1} t_i, \qquad t_i t_{i+1} t_i^{-1} = t_{i+1}^{-1} t_i t_{i+1},$$
$$t_{i+1} t_i^{-1} t_{i+1}^{-1} = t_i^{-1} t_{i+1}^{-1} t_i, \qquad t_i t_{i+1}^{-1} t_i^{-1} = t_{i+1}^{-1} t_i^{-1} t_{i+1}, \qquad t_i^{-1} t_{i+1}^{-1} t_i^{-1} = t_{i+1}^{-1} t_i^{-1} t_{i+1}^{-1}.$$

Thanks to these identities, all sixteen terms of the difference of triple products of the $y_i$ cancel pairwise, but four:

$$y_i(u)y_{i+1}(v)y_i(w) - y_{i+1}(w)y_i(v)y_{i+1}(u) \tag{57}$$
$$= uv^{-1}w(t_i^{-1}t_{i+1}t_i^{-1} - t_{i+1}^{-1}t_it_{i+1}^{-1}) - u^{-1}vw^{-1}(t_it_{i+1}^{-1}t_i - t_{i+1}t_i^{-1}t_{i+1}).$$

Moreover it is easily verified that

$$(t_i^{-1}t_{i+1}t_i^{-1} - t_{i+1}^{-1}t_it_{i+1}^{-1}) = q(t_it_{i+1}^{-1}t_i - t_{i+1}t_i^{-1}t_{i+1}). \tag{58}$$

So the difference (57) will be zero if $quv^{-1}w = u^{-1}vw^{-1}$. This is easily achieved with the following definition of the $x_i$.

**Proposition 4.2.** *Let $n \geq 2$. The $x_i(q,u)$ defined by*

$$x_i(q,u) = \frac{\sqrt{q}}{u}t_i - \frac{u}{\sqrt{q}}t_i^{-1}, \qquad for\ i < n, \tag{59}$$

*satisfy the three conditions* (54)–(56) *with $\rho(q,u) = ((q^2 + q^{-2}) - (u^2 + u^{-2}))$.*

*Proof.* With the new weights $u \mapsto u' = u/\sqrt{q}$, the relation $qu'v'^{-1}w' = u'^{-1}v'w'^{-1}$ with $w' = v'/u'$ is true and the Yang-Baxter is verified. The other two equations are obtained by expanding the $x_i$. □

The solution $x_i$ of the three conditions in 4.1 is well-known. For example, Section 3 of [7] is devoted to this solution and its relationship with the Temperley-Lieb algebra. (Note that their $\lambda$ and our $q$ is related by $q = e^{i\lambda}$ and their $u$ and ours is also related by an exponential. Their $\beta$ is $q + q^{-1}$ while ours is $-q - q^{-1}$. Finally they consider a larger class of boundary conditions that those above.) However the above discussion shows how the braiding of the Temperley-Lieb category $\widetilde{\mathsf{TL}}$ and integrability of statistical models are intimately related.

# 5 The dilute category $\widetilde{\mathsf{dTL}}$

The dilute Temperley-Lieb algebras $\mathsf{dTL}_n(\beta)$ are a family of algebras defined through diagrams similar to those appearing in the original algebras $\mathsf{TL}_n(\beta)$. This family can be cast into a category $\widetilde{\mathsf{dTL}}$ similar to the category $\widetilde{\mathsf{TL}}$ introduced in 2.2. This new category can also be given a braided structure. This section introduces this structure and discusses the relationship between the braiding on $\widetilde{\mathsf{dTL}}$ and the integrability of dilute statistical models.

We start by giving the definition of the category $\widetilde{\mathsf{dTL}}$, while recalling the definitions of the algebras $\mathsf{dTL}_n$ themselves. (See [18] for further details on the dilute family.) The objects of the *dilute Temperley-Lieb category* $\widetilde{\mathsf{dTL}}$ are the non-negative integers. The morphisms between

two integers $n$ and $m$ are defined as linear combinations of *dilute* $(n, m)$-diagrams. These dilute diagrams are defined in the same way as the $(n, m)$-diagrams appearing in $\widetilde{\mathsf{TL}}$ except that nodes on either sides of the diagrams are now allowed to be free of strings; a node without a string is called a *vacancy*. For example, the following are all acceptable dilute diagrams:

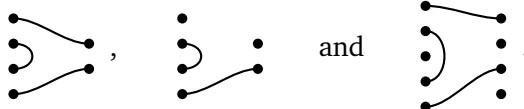

The first two are elements of $\mathrm{Hom}(2, 4)$ and the last of $\mathrm{Hom}(4, 5)$. Composition of morphisms is defined by extending bilinearly the following composition rule. For $b$ and $c$ dilute $(m, n)$- and $(k, m)$-diagrams, the composition $b \circ c$ is a dilute $(k, n)$-diagram defined by first drawing $b$ on the left of $c$, identifying the points on the $m$ neighbouring sites, joining the strings that meets there, and then removing the points on this side. If a string is closed in this process, it is removed and the diagram obtained is multiplied by $\beta = -q - q^{-1} \in \mathbb{C}$. If a string is attached to only one of its extremities (because it was joined to a vacancy during the composition), the result $b \circ c$ is the zero morphism[5]. Here are few examples of these compositions. In the first $b \in \mathrm{Hom}(4, 5), c \in \mathrm{Hom}(2, 4)$ and $b \circ c \in \mathrm{Hom}(2, 5)$.

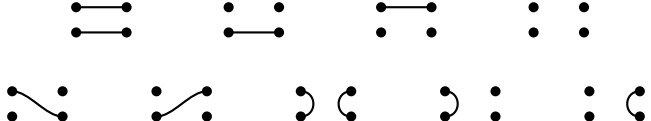

For a stricly positive integer $n$, the algebra $\mathsf{dTL}_n(\beta)$ is identified to the set $\mathrm{Hom}(n, n)$ with the product being the composition just defined.

Endowing this category with a monoidal structure is a straigthforward generalisation of the one on $\widetilde{\mathsf{TL}}$. Again define $n \otimes m \equiv n + m$. For morphisms, if $a$ and $b$ are dilute $(n, m)$- and $(r, s)$-diagrams, define their tensor product $a \otimes b$ as the $(n + r, m + s)$-diagram obtained by simply putting $a$ on top of $b$, as in $\widetilde{\mathsf{TL}}$, and extend this bilinearly to all morphisms. With the associator and the unitors as identities, this tensor product makes $\widetilde{\mathsf{dTL}}$ into a strict monoidal category.

The commutor for $\widetilde{\mathsf{dTL}}$ is obtained similarly to that of $\widetilde{\mathsf{TL}}$. We only outline the computation of $\eta_{1,1}$. The space $\mathrm{End}\, 2$ is spanned by the following 9 diagrams:

and the elementary commutor $\eta_{1,1}$ is a linear combination of these nine diagrams. A line •--• in any diagram will mean the sum of two diagrams, the first with a straight line between the two nodes, the second with nothing between these nodes that are then vacancies. The identity $1_1 \in \mathrm{Hom}(1, 1)$ is thus such a dashed line and the identity in $\mathrm{End}\, 2$

$$1_2 = \begin{smallmatrix} \bullet\text{--}\bullet \\ \bullet\text{--}\bullet \end{smallmatrix}$$

is the sum of the first four of the nine diagrams above. Four of the coefficients of $\eta_{1,1}$ are easily set to zero by the following requirements:

$$\eta_{1,1}\, \begin{smallmatrix} \bullet\text{—}\bullet \\ \bullet\text{--}\bullet \end{smallmatrix} = \begin{smallmatrix} \bullet\text{--}\bullet \\ \bullet\text{—}\bullet \end{smallmatrix}\, \eta_{1,1} \qquad \eta_{1,1}\, \begin{smallmatrix} \bullet\text{--}\bullet \\ \bullet\text{—}\bullet \end{smallmatrix} = \begin{smallmatrix} \bullet\text{—}\bullet \\ \bullet\text{--}\bullet \end{smallmatrix}\, \eta_{1,1}$$

---

[5]Note that the case of a string ending at a vacancy can also be resolved by simply removing it and replacing its two ends by vacancies. This then yields a different structure, the planar rook algebras (see, for example, [19]).

and

$$\eta_{1,1} \begin{smallmatrix}\bullet&&\bullet\\&&\\\bullet&--&\bullet\end{smallmatrix} = \begin{smallmatrix}\bullet&--&\bullet\\&&\\\bullet&&\bullet\end{smallmatrix} \eta_{1,1} \qquad \eta_{1,1} \begin{smallmatrix}\bullet&--&\bullet\\&&\\\bullet&&\bullet\end{smallmatrix} = \begin{smallmatrix}\bullet&&\bullet\\&&\\\bullet&--&\bullet\end{smallmatrix} \eta_{1,1} \,.$$

The commutor $\eta_{1,1}$ is thus found to be a sum of the remaining five diagrams:

$$\eta_{1,1} = a_1 \, \rule{0.5cm}{0pt} + a_2 \, \rule{0.5cm}{0pt} + a_3 \, \rule{0.5cm}{0pt} + a_4 \, \rule{0.5cm}{0pt} + a_5 \, \rule{0.5cm}{0pt} \,.$$

As in 2.2, we define $t_i(n) \equiv 1_{i-1} \otimes \eta_{1,1} \otimes 1_{n-i-1} \in \mathrm{Hom}(n,n)$ and write $\eta_{1,2} = t_2 t_1$ and $\eta_{2,1} = t_1 t_2$. The conditions are then

$$\eta_{1,2}(a \otimes b) = (b \otimes a)\eta_{1,2} \qquad \text{and} \qquad \eta_{2,1}(b \otimes a) = (a \otimes b)\eta_{2,1}$$

for all $a \in \mathrm{dTL}_1$ and $b \in \mathrm{dTL}_2$. Choosing $a$ as $1_1$ and $b$ as one of the following ones

gives the algebraic equations

$$a_1^2 + a_1 a_5 \beta + a_5^2 = 0 \qquad \text{and} \qquad a_2^2 = a_3^2 = a_4^2 = a_1 a_5.$$

Finally the conditions $\eta_{1,1}(a \otimes b) = (b \otimes a)\eta_{1,0} = b \otimes a$ and $\eta_{1,1}(b \otimes a) = (a \otimes b)\eta_{0,1} = a \otimes b$ with $a \in \mathrm{dTL}_1$ and $b \in \mathrm{Hom}(0,1)$ give $a_2 = a_3 = a_4 = 1$. The first equation above becomes $a_1^2 + \beta + a_1^{-1} = 0$ whose solutions are $a_1 = \pm q^{\pm\frac{1}{2}}$, the two $\pm$ being independent. We shall choose both upper signs. This determines completely $\eta_{1,1}$ and it is possible to check that all other conditions on $\eta_{1,1}$, $\eta_{1,2} = t_2 t_1$ and $\eta_{2,1} = t_1 t_2$ are satisfied.

**Proposition 5.1.** *The category* $\widetilde{\mathrm{dTL}}$ *is braided for a commutor with components* $\eta_{r,s}$ *given by* (23), $t_i(n) \equiv 1_{i-1} \otimes \eta_{1,1} \otimes 1_{n-i-1}$ *and* $\eta_{1,1}$ *and* $\eta_{1,1}^{-1}$ *now given by*

$$\eta_{1,1} = q^{\frac{1}{2}} \, \rule{0.5cm}{0pt} + q^{-\frac{1}{2}} \, \rule{0.5cm}{0pt} + \rule{0.5cm}{0pt} + \rule{0.5cm}{0pt} + \rule{0.5cm}{0pt} \tag{60}$$

*and*

$$\eta_{1,1}^{-1} = q^{-\frac{1}{2}} \, \rule{0.5cm}{0pt} + q^{\frac{1}{2}} \, \rule{0.5cm}{0pt} + \rule{0.5cm}{0pt} + \rule{0.5cm}{0pt} + \rule{0.5cm}{0pt} \tag{61}$$

It follows that a disjoint module category $\mathrm{Mod}_{\widetilde{\mathrm{dTL}}}$ can be defined along the lines introduced in 3.1 and that it is also braided.

In the case of the original Temperley-Lieb algebras, the construction of $x_i(q,u)$ satisfying the three conditions (54)–(56) rested on the identities (16) and (58). These can be shown to be satisfied by the $t_i$ defined with $\eta_{1,1}$ in (60). The elementary braiding (60) thus leads again to the following non-trivial solution of the Yang-Baxter equation (54):

$$x_i(q,u) = \frac{\sqrt{q}}{u} t_i - \frac{u}{\sqrt{q}} t_i^{-1}, \qquad \text{for } i < n,$$

where $\eta_{1,1}$ is now given by (60) and the $x_i$ are understood as elements of $\mathrm{Hom}(n,n)$. Does this solution also satisfy the two other conditions (55) and (56)? For the latter, one has first to decide what is to replace the "boundary terms" $(z \otimes z)$. A direct calculation shows that (56) is satisfied by the dilute $x_i$ for only three boundary conditions, namely

, and .

Finally the dilute $x_i$ does not satisfy the inversion relation (55):

$$x_i(q,u)x_i(q,u^{-1}) = ((q+q^{-1})-(u^2+u^{-2}))1_2 + (q^2-q-q^{-1}+q^{-2})\;\rule[0.3ex]{1.2em}{0.6pt}\;.$$

While this non-trivial solution only partially satisfies (54)–(56), there is another one that solves the three conditions. It uses Boltzmann weights discovered by Izergin and Korepin [20], and Nienhuis [21]. It is

$$\hat{x}_i(q,u) = u^{-2}\cdot y_+ + u^{-1}\cdot w_+ + z + u\cdot w_- + u^2\cdot y_-,  \tag{62}$$

where

$$y_\pm = \frac{-q^{\pm\frac{3}{4}}}{(q^{\frac{1}{2}}-q^{-\frac{1}{2}})(q^{\frac{3}{4}}-q^{-\frac{3}{4}})}\Big( -q^{\pm\frac{1}{2}}\;\rule[0.3ex]{1.2em}{0.6pt}\; -q^{\mp\frac{1}{2}}\;\text{⌒}\; \text{⌣} +\diagdown +\diagup + \;\;\Big),$$

$$w_\pm = \frac{\pm 1}{(q^{\frac{3}{4}}-q^{-\frac{3}{4}})}\Big( q^{\pm\frac{3}{4}}(\;\cdot\!\!-\!\!\cdot\; + \;\cdot\!\!-\!\!\cdot\;) - (\;\text{⌒}\;\;+\;\;\text{⌣}\;) \Big),$$

$$z = \frac{1}{(q^{\frac{1}{4}}-q^{-\frac{1}{4}})(q^{\frac{3}{4}}-q^{-\frac{3}{4}})}\Big( (q-1+q^{-1})\;\;\;\; - (\;\rule[0.3ex]{1.2em}{0.6pt}\; + \;\text{⌒}\;\;\text{⌣}\;)$$
$$+ (q^{\frac{1}{2}}-1+q^{-\frac{1}{2}})(\;\diagdown + \diagup\;)\Big).$$

Like $x_i$, this $\hat{x}_i$ solves the Yang-Baxter equation. But it also satisfies the inversion relation (with a new function $\hat{\rho}$) and the boundary Yang-Baxter equation with particular boundary conditions [22, 23]. Notice that, up to a global factor, the $y_\pm$ are the commutors $\eta_{1,1}^{\pm 1}$ that would have been obtained if $a_1$ would have been chosen as $-q^{\frac{1}{2}}$. The other three terms $w_+, w_-$ and $z$ are however completely different. It is not clear whether the integrable model it defines is related to a braiding for a different bifunctor $-\otimes' -$.

## 6 Conclusion

The main results of this article lie in Sections 2 and 3. In Section 2, the category $\widetilde{\mathsf{TL}}$ was given the structure of a braided category. Even though several of the functors and natural morphisms were already known, casting them in $\widetilde{\mathsf{TL}}$ pursues Graham and Lehrer's goal of understanding the family of Temperley-Lieb algebras as a whole. This goal highlights properties of the family that are shared by all $\mathsf{TL}_n$, independently of $n$. It is also very natural physically speaking as the continuum limit of the lattice models defined using the $\mathsf{TL}$ family is often their *raison d'être*. Section 3 introduced the category $\mathrm{Mod}_{\widetilde{\mathsf{TL}}}$ of modules over $\widetilde{\mathsf{TL}}$ and used the functors and natural morphisms of $\widetilde{\mathsf{TL}}$ to induce the structure of a ribbon category on $\mathrm{Mod}_{\widetilde{\mathsf{TL}}}$ when $q$ is generic. The tools developed showed how non-trivial can the monodromy be, even for the finite associative $\mathsf{TL}$ algebras.

Section 3 also explained that rigidity cannot be implemented straightforwardly on $\mathrm{Mod}_{\widetilde{\mathsf{TL}}}$ when $q$ is a root of unity. The subcategory $\mathrm{Mod}_{\widetilde{\mathsf{TL}}}^{\mathrm{proj}}$ of projective modules might satisfy the axioms (47) and (48), but it fails to be abelian. Another possibility would be to consider the subcategory of irreducible modules $\mathsf{I}_{n,k}$ with $k$ on the left of the first critical line. Theorem 6.11 of [4] shows that $\times_f$ is closed on this subset of modules. But a more central question is what are "approriate" weaker forms of rigidity for $\mathrm{Mod}_{\widetilde{\mathsf{TL}}}$ at $q$ a root of unity.

The question also arises of the existence of a commutor for other family of algebras and its eventual link to integrable models defined using them. 5 showed that the "elementary braiding $\eta_{1,1}$" does not reproduce the transfer matrix defining dilute loop models. Is it possible to

understand better the link between this $\eta_{1,1}$ and the integrability of the dilute models? There are other algebras physically relevant in statistical physics, for example the one-boundary TL family (also known as the *blob algebra* [24]) and the affine (periodic) TL family. It is not known whether one can define a fusion product between their modules or even make a braided category out of their link diagrams.

# A  The central element $c_n$

We gather in this appendix the properties of the central element $c_n \in \mathsf{TL}_n$ together with their proofs. Some of these results are to be found in [13].

The two elements of $\mathsf{TL}_n$

$$\rho_n = t_1 t_2 \dots t_{n-1} \qquad \text{and} \qquad \lambda_n = t_{n-1} \dots t_2 t_1,$$

are invertible since each of the $t_i$ is. Define $e_0$ and $e_n$ as

$$e_n = \rho_n e_{n-1} \rho_n^{-1} \qquad \text{and} \qquad e_0 = \lambda_n e_1 \lambda_n^{-1}.$$

Then the following properties hold.

**Lemma A.1.** *The action by conjugation of $\rho_n$ and $\lambda_n$ on elements of $\mathsf{TL}_n$ amounts to right and left cyclic translations:*

$$
\begin{aligned}
\rho_n e_i \rho_n^{-1} &= e_{i+1}, & 1 \le i \le n-1, & \qquad & \lambda_n e_i \lambda_n^{-1} &= e_{i-1}, & 1 \le i \le n-1, \\
\rho_n e_n \rho_n^{-1} &= e_1, & & & \lambda_n e_0 \lambda_n^{-1} &= e_{n-1}, \\
e_{n-1} e_n e_{n-1} &= e_{n-1}, & & & e_0 e_1 e_0 &= e_0, \\
e_n e_{n-1} e_n &= e_n, & & & e_1 e_0 e_1 &= e_1, \\
e_n^2 &= \beta e_n, & & & e_0^2 &= \beta e_0.
\end{aligned}
$$

*Proof.* The first relations follow from 2.3. The repeated use of the same identity (15) proves the second:

$$
\begin{aligned}
\rho_n e_n \rho_n^{-1} &= \rho_n^2 e_{n-1} \rho_n^{-2} = (t_1 t_2 \dots t_{n-1})(t_1 t_2 \dots t_{n-1}) e_{n-1} (t_{n-1}^{-1} \dots t_2^{-1} t_1^{-1})(t_{n-1}^{-1} \dots t_2^{-1} t_1^{-1}) \\
&= (t_1 t_2 \dots t_{n-2})(t_1 t_2 \dots t_{n-3})(t_{n-1} t_{n-2}) e_{n-1} (t_{n-2}^{-1} t_{n-1}^{-1})(t_{n-3}^{-1} \dots t_2^{-1} t_1^{-1})(t_{n-2}^{-1} \dots t_2^{-1} t_1^{-1}) \\
&= (t_1 t_2 \dots t_{n-2})(t_1 t_2 \dots t_{n-3}) e_{n-2} (t_{n-1} t_{n-2} t_{n-2}^{-1} t_{n-1}^{-1})(t_{n-3}^{-1} \dots t_2^{-1} t_1^{-1})(t_{n-2}^{-1} \dots t_2^{-1} t_1^{-1}) \\
&= (t_1 t_2 \dots t_{n-2})(t_1 t_2 \dots t_{n-3}) e_{n-2} (t_{n-3}^{-1} \dots t_2^{-1} t_1^{-1})(t_{n-2}^{-1} \dots t_2^{-1} t_1^{-1}) \\
&= \dots = (t_1 t_2)(t_1) e_2 (t_1^{-1})(t_2^{-1} t_1^{-1}) = t_1 e_1 t_1^{-1} = e_1.
\end{aligned}
$$

The cubic relations are straightforward. For example

$$e_n e_{n-1} e_n = \rho_n e_{n-1} (\rho_n^{-1} e_{n-1} \rho_n) e_{n-1} \rho_n^{-1} = \rho_n e_{n-1} e_{n-2} e_{n-1} \rho_n^{-1} = \rho_n e_{n-1} \rho_n^{-1} = e_n.$$

Finally the square of $e_n$ is obtained by $e_n^2 = (\rho_n e_{n-1} \rho_n^{-1})^2 = \rho_n (\beta e_{n-1}) \rho_n^{-1} = \beta e_n$. $\qquad\square$

The definition and properties of the elements $c_n \in \mathsf{TL}_n$ that are used in the study of the twist $\theta$ are contained in the next Proposition. The statement refers to the standard modules $\mathsf{S}_{n,k}$, $0 \le k \le n$ with $k \equiv n \bmod 2$, over $\mathsf{TL}_n$. These were defined in 3.1. Again a basis for $\mathsf{S}_{n,k}$ can be chosen to be the $(n,k)$-diagrams in $\mathrm{Hom}(k,n)$ with $k$ through lines. (More information on these modules can be found in [13–15, 25]).

**Proposition A.2.** *The elements $c_n \overset{\text{def}}{=} q^{3n/2}\rho_n^n$ and $d_n \overset{\text{def}}{=} q^{3n/2}\lambda_n^n \in \mathsf{TL}_n$ satisfy the following properties:*

(a) *$c_n$ and $d_n$ are invertible central elements of $\mathsf{TL}_n$;*

(b) *$c_n$ and $d_n$ act on the standard modules $\mathsf{S}_{n,k}$ as $\gamma_{n,k} = q^{\frac{1}{2}k(k+2)}$ times the identiy;*

(c) *if $q$ is not a root of unity, then the powers $\{1_n, c_n, c_n^{\,2}, \ldots, c_n^{\lfloor n/2 \rfloor}\}$ of $c_n$ form a basis of the center of $\mathsf{TL}_n$;*

(d) *$c_n = d_n$.*

The factor $q^{3n/2}$ is the definition of $c_n$ and $d_n$ will be useful in the identification of the component $\theta_n$ of the natural isomorphism $\theta$ with $c_n$.

*Proof.* Statement (a) follows from the invertibility of the $t_i$'s and the cyclic property proved in the previous Lemma. For (b), note that $c_n$ and $d_n$ being central, they define endomorphisms of the standard modules $\mathsf{S}_{n,k}$ by left multiplication. Since $\mathrm{Hom}(\mathsf{S}_{n,k}, \mathsf{S}_{n,k}) \simeq \mathbb{C}$, these morphisms must be multiples of the identity. Let $\gamma_{n,k}$ be the multiple for the morphism defined by $c_n$. Then, on $\mathsf{S}_{n,k}$, $(q^{-3n/2}\gamma_{n,k})^{\dim \mathsf{S}_{n,k}} = \det(q^{-3n/2}c_n) = (\det \rho_n)^n$. But all the $t_i$ are conjugate and $\det \rho_n = \det(t_1 t_2 \ldots t_{n-1}) = (\det t_1)^{n-1}$. Thus

$$(q^{-3n/2}\gamma_{n,k})^{\dim \mathsf{S}_{n,k}} = (\det t_1)^{n(n-1)}.$$

To compute the determinant of $t_1$, choose a basis where $(n,k)$-diagrams with an arc between position 1 and 2 appear first. There are $\dim \mathsf{S}_{n-2,k}$ such diagrams and $e_1$ acts on them as $\beta$ times the identity. Moreover, on any other $(n,k)$-diagrams, $e_1$ acts either as zero or gives a diagram with an arc between 1 and 2. Therefore $e_1$ takes the form

$$e_1 = \begin{pmatrix} \beta I & ? \\ 0 & 0 \end{pmatrix}$$

in this basis. (Here $I$ is the identity matrix of size $\dim \mathsf{S}_{n-2,k} \times \dim \mathsf{S}_{n-2,k}$.) The matrix representing $t_1$ is thus

$$q^{\frac{1}{2}} \begin{pmatrix} (1+q^{-1}\beta)I & ? \\ 0 & I' \end{pmatrix}$$

and (

$$\det t_1 = q^{\frac{1}{2}\dim \mathsf{S}_{n,k}}(-q^{-2})^{\dim \mathsf{S}_{n-2,k}}.$$

The dimension of the standard module $\mathsf{S}_{n,k}$ is $\binom{n}{(n-k)/2} - \binom{n}{(n-k)/2-1}$ and a direct computation shows that $n(n-1)\dim \mathsf{S}_{n-2,k} = \frac{1}{4}(n-k)(n+k+2)\dim \mathsf{S}_{n,k}$ which gives

$$\gamma_{n,k} = \omega \times q^{\frac{1}{2}k(k+2)}$$

where $\omega$ is a root of unity such that $\omega^{\dim \mathsf{S}_{n,k}} = 1$. The central element $c_n$ is a Laurent polynomial in $q$. (There are $n(n-1)$ factors $t_i = q^{\frac{1}{2}}(1_n + q^{-1}e_i)$ in $c_n = (t_1 t_2 \ldots t_{n-1})^n$ and their factors $q^{\frac{1}{2}}$ are thus in even numbers.) The eigenvalue $\gamma_{n,k}$ will thus be continuous, except maybe at $q = 0$ or $\infty$. The root $\omega$ must thus be constant. At $q = 1$, the Temperley-Lieb algebra $\mathsf{TL}_n(\beta = 2)$ is known to be a quotient of the group algebra of the symmetric group $\mathsf{S}_n$ and the elements $t_i$ are then the transposition $(i, i+1)$. Thus $\rho_n$ is the permutation $(1, 2, \ldots, n)$ and the central element $c_n$ is the identity permutation. Hence $\gamma_{n,k} = 1$ at $q = 1$ and the only possible choice for $\omega$ is 1.

If $q$ is not a root a unity, then $\mathsf{TL}_n(\beta = -q - q^{-1})$ is semisimple and is known to have $\lfloor n/2 \rfloor + 1$ inequivalent irreducible modules. By Wedderburn's theorem the algebra then decomposes into two-sided ideals, one for each inequivalent irreducible, and each ideal is isomorphic to the algebra of $d \times d$-matrices, if the corresponding irreducible is of dimension $d$. The dimension of the center of $\mathsf{TL}_n$ at such a $q$ is thus $\lfloor n/2 \rfloor + 1$. Moreover the eigenvalues $\gamma_{n,k}, 0 \leq k \leq n$ with $k \equiv n \bmod 2$, computed in (b) are distinct when $q$ is not a root of unity. Therefore the minimal polynomial of $c_n$ in the (faithful) regular representation has degree $\lfloor n/2 \rfloor + 1$ and the powers $\{1_n, c_n, c_n{}^2, \ldots, c_n{}^{\lfloor n/2 \rfloor}\}$ are linearly independent. They must form a basis of the center of $\mathsf{TL}_n$.

The computation of the determinant of $d_n$ follows the same line than that of $c_n$. Since $c_n$ and $d_n$ contain the same number of generators $t_i$, their eigenvalues on the standard modules thus coincide. When $q$ is not a root of unity, these eigenvalues completely determine the linear decomposition in the basis obtained in (c) and $d_n$ and $c_n$ must be equal. By continuity, they must also be equal at roots of unity. □

When $q$ is a root of unity, the algebra $\mathsf{TL}_n(\beta = -q - q^{-1})$ is in general not semisimple. A list of its indecomposable modules is known ( [15], see also [25]) and the only indecomposable modules $\mathsf{M}$ whose endomorphism groups $\mathrm{Hom}(\mathsf{M}, \mathsf{M})$ are larger than $\mathbb{C}$ are the projective modules $\mathsf{P}_{n,k}$ that have three or four composition factors. Statement (b), above, can be extended to all others.

**Corollary A.3.** *Let* $\mathsf{M}$ *be a module over* $\mathsf{TL}_n$ *such that* $\mathrm{Hom}(\mathsf{M}, \mathsf{M}) \simeq \mathbb{C}$ *and let* $\mathsf{I}_{n,k}$ *be one of its composition factors.*[6] *Then* $c_n$ *acts on* $\mathsf{M}$ *as* $\gamma_{n,k} \cdot \mathrm{id}$.

If $\mathsf{M}$ is an indecomposable projective such that $\mathrm{Hom}(\mathsf{M}, \mathsf{M}) \simeq \mathbb{C}^2$, then $c_n$ will still have a single eigenvalue on $\mathsf{M}$, but it might not be a multiple of the identity. Such possibility will occur in the examples of the monodromy $\eta_{\mathsf{N},\mathsf{M}} \circ \eta_{\mathsf{M},\mathsf{N}}$ given in 3.3.

# Acknowledgements

We thank Thomas Creutzig, David Ridout and Ingo Runkel for helpful discussions. The authors also thank the referee for his/her constructive comments; they improved the presentation. This work was done while JB was holding scholarships from Fonds de recherche Nature et Technologies (Québec) and from the Faculté des études supérieures et postdoctorales de l'Université de Montréal, and is supported by the European Research Council (advanced grant NuQFT). YSA holds a grant from the Canadian Natural Sciences and Engineering Research Council. This support is gratefully acknowledged.

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
