# Peer review of "Fusion and monodromy in the Temperley-Lieb category"

_SciPost Physics, doi:SciPost Phys. 5, 041 (2018)_

## Round 2 · Referee Report · Anonymous (Referee 1) · 2018-9-6

Strengths

1- Beautifully written.

2- Strategies for lengthy steps and derivations explained clearly.

Weaknesses

1- None that I can think of.

Report

In this paper, authors consider a category (denoted $\widetilde{\mathsf{TL}}$) which is designed to capture all the Temperley-Lieb algebras in a single framework.
They endow this category with the structure of a braided tensor category with a twist.
Then they introduce a category of modules for $\widetilde{\mathsf{TL}}$ and
provide a ribbon structure on this category.
These are the main results of the paper.
Analogous ribbon structures are quite important from many points of view, especially they appear frequently in many CFT considerations where fusion is intimately related to the operator product expansions.
Authors also discuss relations to integrability of statistical models.

The paper is beautifully written and is a delight to read.
The results are deep, interesting and new yet the paper could be read by a graduate student with a rudimentary knowledge of category theory and basic algebra.
The steps are explained clearly and enough motivation is provided.

I have only found one mathematical issue with the paper (point 1 below). I wholeheartedly recommend its publication after this issue is resolved.
Further, I am requesting a few other changes, almost all of them are completely minor, such as typos etc.

Requested changes

  1. The main mathematical issue is thus: Let $\mathcal{C}$ be a rigid tensor category. Then, given any morphism $f:X\rightarrow Y$ in $\mathcal{C}$, there is a left and right dual to $f$, one of which is given by:

    $$f^\ast: Y^\ast \xrightarrow{1\otimes i_X} Y^\ast\otimes(X\otimes X^\ast) \xrightarrow{1\otimes (f\otimes 1)} Y^\ast\otimes(Y\otimes X^\ast) \xrightarrow{\mathrm{\alpha^{-1}}} (Y^\ast\otimes Y)\otimes X^\ast\xrightarrow{e_Y\otimes 1} X^\ast,$$
    and the other one is similar. For a twist to yield a ribbon structure it is required to be compatible with the rigidity, i.e., $(\theta_X)^\ast = \theta_{X^\ast}$ for all objects $X$ in $\mathcal{C}$ (see Definition 8.10.1 from the book by Etingof, Gelaki, Nikshych, Ostrik). This condition needs to be checked for Proposition 3.7. It may be an easy verifaction, but it must be carried out.

  2. Is the category $\widetilde{\mathsf{TL}}$ abelian? Please make a remark accordingly. The rigidity of $\mathrm{Mod}_{\widetilde{{\mathsf{TL}}}}$ is discussed in Section 3. Is $\widetilde{\mathsf{TL}}$ also rigid? Ribbon?

  3. The definition of natural (iso)morphisms is recalled in the opening paragraph of Section 2.2. It is better to move this in Section 2.1, where natural isomorphisms first enter, in the definition of the monoidal category.

  4. Last line of Lemma 2.2: it should be $\prod_{i=s}^1t_i$. The $t_i$ part is missing.

  5. For the braiding in $\widetilde{\mathsf{TL}}$. Once one has checked that $\eta_{\cdot,\cdot}$ provides the braiding, is it beneficial to introduce a diagram for it, by representing it with crossing lines (either under or over crossing, make a choice)? With these diagrams, truth of some results, esp. Lemmas 2.4 - 2.5, 2.9 etc., may also be explained pictorially: morphisms with isotopic diagrams are equal. I haven't checked if such diagrams are incompatible with anything, and it may be potentially dangerous to introduce them; but it will be good if the authors could ponder on this point. They may chose to not make any changes.

  6. On the fourth line on page 12, there are too many closing parentheses.

  7. In the very last line of proof of Proposition 2.10, it appears that $\mathsf{S}_{n,0}$ are used, but they are defined only in the later section.

  8. Near equation (29), the action of the functor on morphisms should also be defined. Similarly for equation (31).

  9. For the footnote 1 on page 14, perhaps it is better to append another clause, ``as the latter contain only semi-simple modules, among other requirements.''

  10. On page 17, 3rd line, ``often the left and right partners coincide''. I think they are isomorphic if the category is ribbon.

  11. In proposition 3.10, perhaps it is better to avoid using the letter $\alpha$ for that constant, since $\alpha$ already denotes associativity and similarly, later on page 21, maybe avoid using $\eta$ $(=\eta_{\mathsf{TL}2,\mathsf{TL}_2} \circ \eta)$}_2,\mathsf{TL}_2 to denote monodromy to avoid confusion with braiding.

  • validity: top
  • significance: high
  • originality: high
  • clarity: top
  • formatting: perfect
  • grammar: perfect

Author:  Yvan Saint-Aubin  on 2018-10-03  [id 325]

(in reply to Report 1 on 2018-09-06)
Category:
correction

First, we would like to thanks the referee for his or her many constructive comments. Several have brought improvement to our text and a word of acknowledgement has been added at the end of the paper.

We reply by referring to the numbered comments. We have already made changes to our text, but SciPost seems to prefer for the new version not to be sent.

  1. Good point. We added these verifications just before proposition 3.7; also added a definition for the duals of twists after proposition 3.6.

  2. The category $\mathsf{\widetilde{TL}}$ is not abelian since it is not additive. It might be possible to endow the category with biproducts and ``fix" this but we haven't explored this issue. The most naive case $n \oplus m = n+m$ does not work when one of $n$ or $m$ is odd. We've added a footnote in definition 1 to outline the fact that the tensor product is not a direct sum.

It is however both rigid and ribbon and we've added remarks in the text where rigidity and ribbons are introduced.

  1. Moved the definition to a footnote in section 2.1 where the natural morphisms are first mentioned, just before definition 1.

  2. Fixed.

  3. Such diagrams with crossing can indeed be introduced, but they do not form a faithful presentation of the rings of morphisms. Indeed we provide an example (new equation (13)) where a Reidermeister move of type I links two isotopic diagrams that are only equal up to a constant.

That being said, we agree that diagrammatic representations of some of these relations are useful and we have added numerous diagrams in section 2, to illustrate the various lemmas.

  1. Fixed (now middle of page 13).

  2. You are right (unfortunately...). There is no elegant way to solve this. We chose to warn the reader with a footnote (bottom of page 13) at the beginning of the proof.

  3. There was a short description of the action of the functors on morphism in the paragraphs near equation (29) (now equation (36)) but we have added an explicit definition in equation (37).

We've also added a definition of the action of the fusion bifunctor (equation (37)) right after what was equation (31).

  1. Added a comment in the footnote (now on page 15) to mention that fusion categories have other structures as well.

  2. Actually they are isomorphic if the category is braided; added a comment on page 18, after equation (48).

  3. Changed the $\alpha$ in 3.10 to $\mu$, and the monodromy to $\delta$.

---

## Round 3 · Referee Report · Anonymous · 2018-10-16

Strengths

Same as in the previous report.

Weaknesses

Same as in the previous report.

Report

The authors have incorporated all the requested changes.
I recommend that this paper be accepted.

Requested changes

None.

---

## Editorial Decision

published